# Interplay of p23 with FKBP51 and their chaperone complex in regulating tau aggregation

Pijush Chakraborty [1] & Markus Zweckstetter [1,2] ✉

The pathological deposition of tau and amyloid-beta into insoluble amyloid fibrils are pathological hallmarks of Alzheimer's disease. Molecular chaperones are important cellular factors contributing to the regulation of tau misfolding and aggregation. Here we reveal an Hsp90-independent mechanism by which the co-chaperone p23 as well as a molecular complex formed by two co-chaperones, p23 and FKBP51, modulates tau aggregation. Integrating NMR spectroscopy, SAXS, molecular docking, and site-directed mutagenesis we reveal the structural basis of the p23-FKBP51 complex. We show that p23 specifically recognizes the TPR domain of FKBP51 and interacts with tau through its C-terminal disordered tail. We further show that the p23-FKBP51 complex binds tau to form a dynamic p23-FKBP51-tau trimeric complex that delays tau aggregation and thus may counteract Hsp90-FKBP51 mediated toxicity. Taken together, our findings reveal a co-chaperone mediated Hsp90-independent chaperoning of tau protein.

Cellular homeostasis by molecular chaperones is indispensable for the survival of long-lived neurons. Neurodegenerative diseases, including tauopathies, are associated with an imbalanced protein load[1]. Tauopathies are a class of neurodegenerative disorders that are characterized by the deposition of tau protein as insoluble aggregates in neurons[2]. Hsp90 is a key molecular chaperone that along with the help of several other co-chaperones stabilizes tau and maintains tau levels[1,3] (Fig. 1a). Direct modulation of Hsp90 by small molecules or other modalities may thus be a pertinent approach to develop therapies for tauopathies and other neurodegenerative disorders[4–7]. However, as Hsp90 interacts with a diverse set of client proteins including steroid receptors and kinases[8–11], direct regulation of Hsp90 activity might be detrimental. Co-chaperones of Hsp90 that target a specific subgroup of proteins, e.g., intrinsically disordered proteins (IDPs), may thus be more viable drug targets for the development of therapies with fewer side effects[1]. For example, the knockdown of the two co-chaperones, p23 and FKBP51, led to a dramatic reduction in total tau levels in cells suggesting a stabilizing effect of these co-chaperones on tau that is independent of Hsp90 activity[1,7]. However, the molecular basis of Hsp90-independent chaperoning of tau by co-chaperones is largely unknown.

p23 is the smallest protein in the Hsp90 machinery and is highly conserved in eukaryotes[12–14]. The human p23 (molecular weight = 19 kDa) is 160 amino acids long and comprises a folded domain (residue 1 to 110) and a large disordered tail (residue 111 to 160)[15]. The folded domain of p23 consists of anti-parallel β-sheets and is similar to the α-crystallin domain of small heat shock proteins[16–18]. The residues present at the beginning of the C-terminal tail of p23 form a conserved helix that interacts with the client binding site of Hsp90[19]. As an Hsp90 co-chaperone, p23 inhibits the ATPase activity of Hsp90, stabilizes its closed conformation, and enhances the client maturation[20–22]. In addition to modulating Hsp90 activity, p23 acts as an independent molecular chaperone and prevents protein aggregation[23–25]. The C-terminal tail of p23 is reported to be essential for its chaperone function[26]. In our recent study, we showed that p23 directly interacts with tau[27]. However, the residues of p23 that are involved in interaction with tau are still unknown.

FKBP51, a 51 kDa molecular chaperone, is highly expressed in neurons[28]. It is a member of the peptidyl-prolyl cis-trans isomerase (PPIase) family of proteins and binds to the immunosuppressive drug FK506[29]. It consists of three folded domains, two FK506-binding

[1]Department for NMR-based Structural Biology, Max Planck Institute for Multidisciplinary Sciences, Göttingen, Germany. [2]German Center for Neurodegenerative Diseases (DZNE), Göttingen, Germany. ✉e-mail: markus.zweckstetter@dzne.de

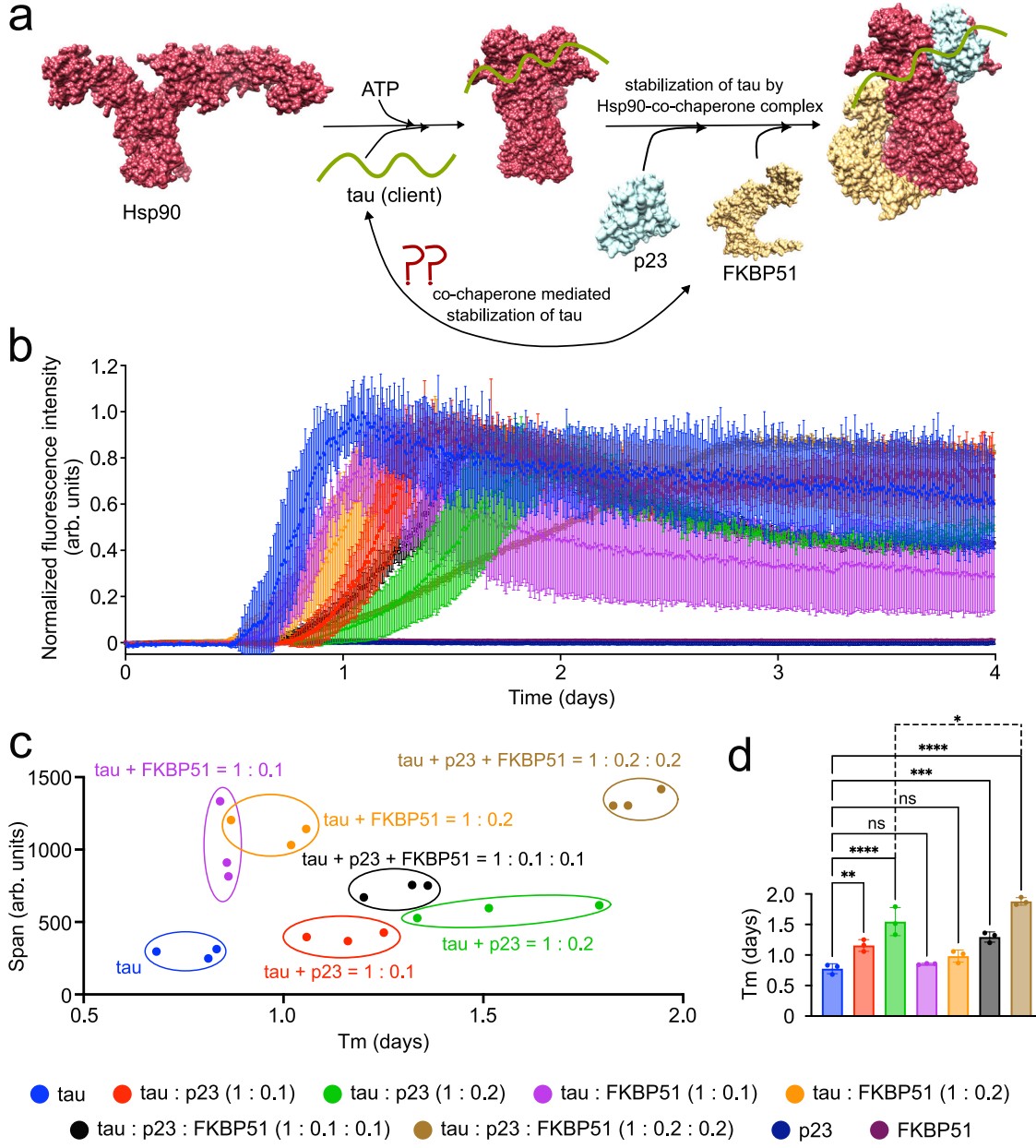

**Fig. 1 | p23-FKBP51 modulates the aggregation of tau. a** Schematic representation of the stabilization of tau by an Hsp90-co-chaperone complex. Hsp90 undergoes ATP-dependent conformational changes, from open to a closed state. Tau can bind to either the open or closed conformation of Hsp90. The client tau may be further stabilized by the Hsp90-FKBP51-p23 complex. PDB ID of open Hsp90 – 2IOQ[58], Hsp90-p23-FKBP51 complex - 7L7I[50]. **b** Aggregation kinetics of 25 μM tau in the absence or presence of different co-chaperones. The fluorescence intensity is normalized. The non-normalized curves are shown in Supplementary Fig. 1. Error bars represent the std of three independently aggregated samples. The center of the error bars represents the average value of three independent samples. **c** ThT-intensity span vs. half time of aggregation (Tm) of tau either in the absence or presence of different co-chaperones. The span is calculated from the non-normalized aggregation curves shown in Supplementary Fig. 1. Error bars represent the std of three independently aggregated samples. **d** Aggregation half-time (Tm) of tau either in the absence or presence of different co-chaperones. Error bars represent the std of three independently aggregated samples. The center of the error bars represents the average value of three independent samples. Statistical analysis between free tau and all other tau-co-chaperone complexes was performed using one-way ANOVA analysis. **p = 0.0061, ***p = 0.0004, ****$p < 0.0001$. Statistical analysis between tau:p23 = 1:0.2 and tau:p23:FKBP51 = 1:0.2:0.2 was performed using one-way ANOVA analysis (*p = 0.0366). Source data are provided as a Source Data file.

domains (FK1 and FK2), and a tetratricopeptide repeat (TPR) domain. The FK1 domain possesses the PPIase activity and the FK2 domain has an ATP binding site. The TPR domain comprises seven α-helices and is the binding site for different heat-shock proteins (Hsps), tubulin, as well as caters to the chaperone function of FKBP51[30]. FKBP51 plays a crucial role in stress response by regulating the activity of glucocorticoid receptors[31]. It also plays a significant role in tau biology by directly interacting with the proline-rich domain and the repeat domain of tau[32]. Tau phosphorylation decreases its interaction with

FKBP51 suggesting varying interaction of the two proteins at different stages of Alzheimer's disease (AD)[32]. The total tau level in the cell increases upon overexpression of FKBP51 which might be due to the impaired ubiquitination of tau[1]. Also, lower levels of tau were detected in FKBP51 knockout mice, and higher levels of FKBP51 were detected in the AD brain[1,33]. FKBP51 is also reported to stabilize microtubules with tau in a reaction that depends on its PPIase activity[1].

Here we reveal an Hsp90-independent modulation of tau aggregation by a molecular complex of p23 and FKBP51 and reveal its

structural basis. Our results support a protective role of the p23-FKBP51-tau complex that may counteract the protoxic activity of the Hsp90-FKBP51-tau interaction.

## Results

### p23 modulates tau aggregation alone as well as in the presence of FKBP51

To gain insight into the role played by p23 and FKBP51 in modulating tau's aggregation, we aggregated the 441-residue human 2N4R tau (further referred to as tau) either in the absence or presence of individual co-chaperones or a mixture of them using our previously developed co-factor-free in vitro aggregation assay[34] (Fig. 1b, Supplementary Fig. 1). In the absence of the co-chaperones, tau started to form amyloid fibrils after ~14 h. In the presence of either 1:0.1 or 1:0.2 molar ratio of tau to FKBP51, the time to reach half of the ThT fluorescence intensity at the plateau (Tm) was unchanged, suggesting that FKBP51 alone does not modulate the kinetics of tau aggregation (Fig. 1c, d). However, the tau samples in the presence of FKBP51 reached a higher ThT intensity than the unmodified tau (Fig. 1c, Supplementary Fig. 1). Furthermore, the tau fibrils generated in the presence of FKBP51 showed enhanced fluorescence than the unmodified tau fibrils upon addition of another amyloid-binding dye curcumin[35,36] (Supplementary Fig. 2). As a comparable amount of tau aggregated both in the presence and absence of FKBP51 (Supplementary Fig. 3), the higher fluorescence intensity in presence of FKBP51 might be attributed to a different structure of the tau fibril or due to the presence of FKBP51 on the fibril surface.

We then studied the effect of p23 on the aggregation of tau. The addition of a sub-stochiometric amount of p23 (tau:p23 = 1:0.1) already delayed the aggregation of tau (Fig. 1c, d, Supplementary Fig. 1). Increasing the amount of p23 (tau:p23 = 1:0.2) further delayed tau's aggregation. However, unlike FKBP51, the presence of p23 had only little influence on the final fluorescence intensity (Fig. 1c, Supplementary Fig. 1 and 2).

To gain further insight into the modulation of tau aggregation in the presence of both p23 and FKBP51, we added both co-chaperones to a final molar ratio of tau: p23: FKBP51 to 1:0.1:0.1 or 1:0.2:0.2. The presence of both co-chaperones attenuated the aggregation of tau to a slightly larger extent than the individual co-chaperones at the comparable concentrations (Fig. 1c, d). Notably, while the final fluorescence intensity was high even at the 1:0.2:0.2 molar ratio (Supplementary Fig. 1 and 2), quantification of the supernatant suggested a comparable amount of aggregated tau (Supplementary Fig. 3). The combined data indicates that p23 plays a crucial role in delaying the aggregation of tau independently as well as in the presence of FKBP51.

### FKBP51 binds to p23

The findings from the in vitro aggregation assay suggest that the combined presence of FKBP51 and p23 delays tau's aggregation to a slightly larger extent than p23 alone. To investigate the molecular basis of this effect, we first recorded two-dimensional (2D) NMR correlation spectra of p23 (Fig. 2a). The $^1$H-$^{15}$N TROSY heteronuclear single quantum coherence (HSQC) spectrum of deuterated, $^{15}$N-labeled p23 was comparable to previously published data[37]. We then added equimolar, two-fold, and three-fold molar excess of unlabeled FKBP51 to the NMR-observable p23. The addition of FKBP51 significantly decreased the signal intensity of residues present in the structured domain (residue 1 to 110) of p23, suggesting an interaction between the two proteins (Fig. 2a, b). Isothermal titration calorimetry suggested a binding affinity of ~67 μM between the two proteins (Supplementary Fig. 4). The strong signal attenuation observed in the HSQC spectra of p23 can be attributed to the increase in molecular weight due to p23-FKBP51 complex formation that leads to enhanced transverse relaxation. We also observed a decrease in the signal intensity of the p23 residues 114

to 130 (Fig. 2b). Notably, these residues (114–130) present in the C-terminal tail of p23 are predicted by AlphaFold2[38] to form an α-helical structure. A similar α-helix near the structured domain of yeast p23/Sba1 was previously reported to interact with the client binding site of Hsp90[19]. Apart from the decrease in signal intensities we also observed chemical shift perturbation (CSP) of the p23 residues 7–12, 53–60, and 80–110 (Fig. 2b). The highest CSP values were observed for residue A94, followed by R88, and K107. Mapping the residues that showed CSP and intensity changes on the three-dimensional structure of p23 revealed the major binding site of FKBP51 near the C-terminal end of the structured domain of p23 (Fig. 2a).

To gain further insights into the role played by the C-terminal tail of p23 to interact with FKBP51, we created a C-terminal deletion mutant of p23 by deleting the terminal 41 residues. The addition of equimolar, two-fold, and five-fold molar excess of unlabeled FKBP51 to the NMR observable truncated p23 (1–119) protein led to the perturbation of the same residues present in the structured domain of p23 as observed in full-length p23 (Fig. 2c). However, the extent of CSP and intensity changes was much lower (Fig. 2b, c). This suggests that although the main binding site of FKBP51 on p23 is the structured domain, the C-terminal tail is also involved in the interaction.

### p23 recognizes the TPR domain of FKBP51

To elucidate the binding site of p23 on FKBP51, we $^{13}$C-labeled the methyl groups of the leucine and valine residues of FKBP51 as these residues are distributed in all three domains (FK1, FK2, and TPR) (Fig. 3a). We then recorded 2D $^1$H-$^{13}$C HMQC spectra of the FKBP51 and transferred the assignments of the methyl groups present in the FK1 and FK2 domain from our previously published study[32] (Fig. 3b). The methyl groups present in the TPR domain of FKBP51 could not be assigned due to limited solubility of this domain when expressed alone. Upon addition of two-fold, five-fold, and ten-fold molar excess of unlabeled p23 to the $^{13}$C-methyl-labeled FKBP51, we observed changes in the cross peaks of several methyl resonances from the TPR domain while the resonances of the FK1 and FK2 domain remained unchanged. The data identify the TPR domain of FKBP51 as binding site of p23 (Fig. 3c, d).

To further confirm the interaction between p23 and the TPR domain of FKBP51, we prepared a construct of FKBP51 comprising only the FK1 and FK2 domains, referred to as FK1-FK2. We then added two-fold and four-fold molar excess of unlabeled FK1-FK2 to the NMR observable p23. Addition of FK1-FK2 did not induce any changes in the signals of p23 (Supplementary Fig. 5) confirming that the FK1 and FK2 domain of FKBP51 are not involved in the interaction with p23.

### Competitive binding of p23 and MEEVD Hsp90 peptide to FKBP51

The Hsp90 C-terminal ends with a highly conserved MEEVD sequence that directs the binding of different TPR-containing co-chaperones (FKBP51, FKBP52, HOP, Cyp40) to Hsp90[39]. The MEEVD peptide interacts with K272, E273, K352, N322, and K329 residues present in the TPR domain of FKBP51 by forming di-carboxylate clamps[40]. To investigate if p23 and MEEVD peptide share the same binding site on FKBP51, we first formed the complex of NMR observable p23 with a two-fold molar excess of unlabeled FKBP51 and recorded the $^1$H-$^{15}$N TROSY HSQC spectra of p23 both in its free and complex form. The addition of FKBP51 leads to the attenuation of most of the signals from the structured domain of p23 (Fig. 4a). The further addition of ten-fold, twenty-fold, and thirty-fold molar excess of MEEVD peptide leads to recovery of ~20%, 40%, and 60% signal intensity from the structured part of p23, respectively, suggesting gradual removal of p23 from the binding site of FKBP51. This indicates that p23 and MEEVD compete to bind with FKBP51 and demonstrates that their binding sites on the TPR domain of FKBP51 are overlapping.

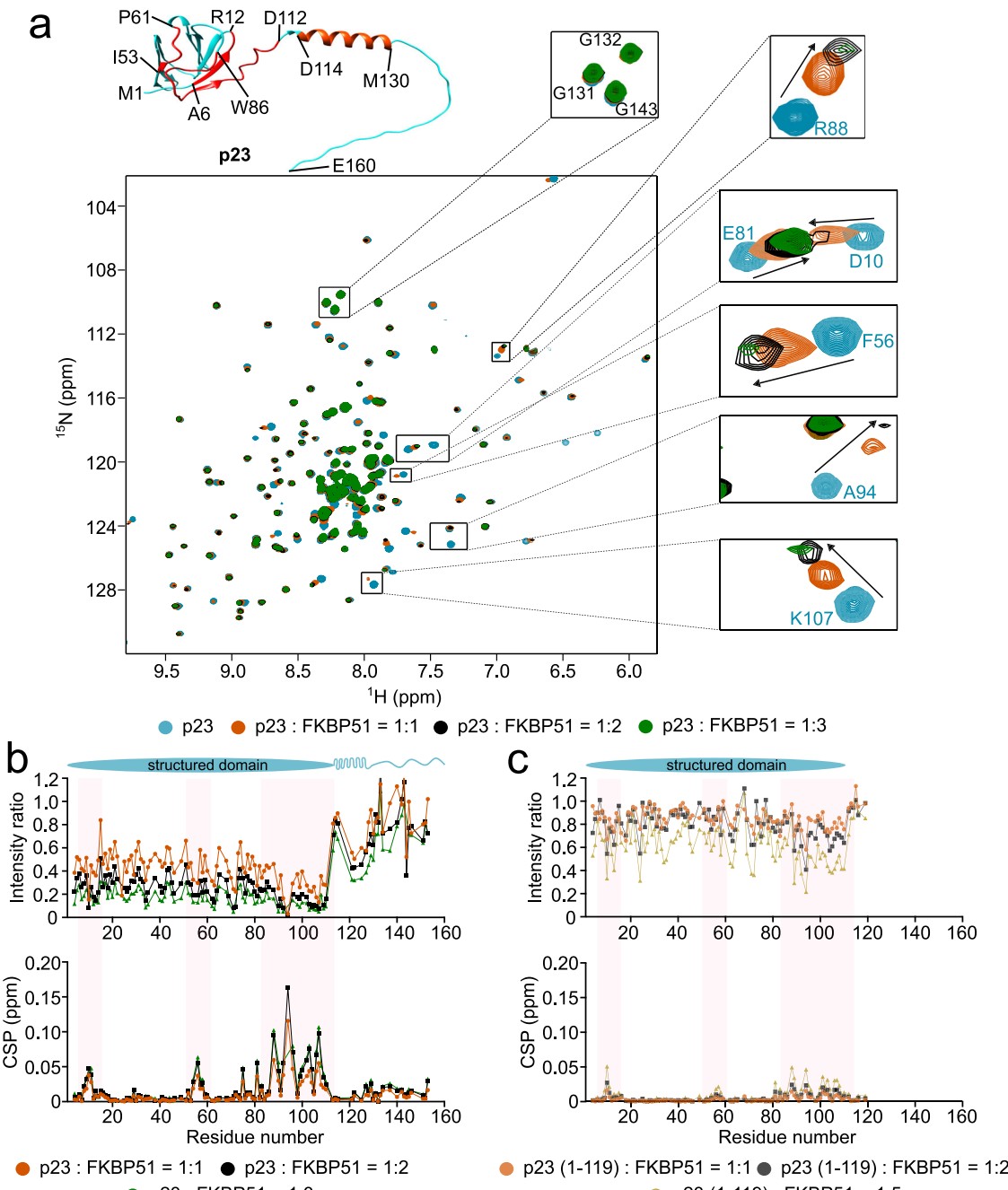

**Fig. 2 | FKBP51 binds to the folded domain of p23. a** 2D $^1$H-$^{15}$N TROSY HSQC spectra of deuterated, $^{15}$N-labeled p23 in the absence (cyan) or presence of equimolar (orange), two-fold (black), and three-fold (green) molar excess of unlabeled FKBP51. An Alphafold2-predicted structure of p23 is shown in the upper left corner. The residues that showed CSP, as well as intensity change, are mapped by red color on the structure. Residues from 114 to 130 in the C-terminal tail of p23 are predicted to form an α-helix. **b, c** Changes in intensities (top) and peak positions (chemical shift

perturbations; CSPs; bottom) of the cross peaks in the TROSY-HSQC spectrum of p23 (**b**) or the truncated p23 (1–119) protein (**c**) upon addition of equimolar (orange), two-fold (black), and three-fold (green)/ five-fold (yellow) molar excess of unlabeled FKBP51. Residues that show strong changes in intensity/CSPs are highlighted with a pink box. The domain diagram of p23 is shown above. Source data are provided as a Source Data file.

To further elucidate the comparable binding site of MEEVD and p23 on FKBP51, we conducted 2D $^1$H-$^{13}$C methyl-TROSY spectra of $^{13}$C-methyl labeled FKBP51 in the absence or presence of five-fold and fifteen-fold molar excesses of MEEVD peptide (Supplementary Fig. 6a). The addition of MEEVD caused perturbations to the same methyl resonances from the TPR domain of FKBP51 as those induced by p23 (Supplementary Fig. 6). However, the majority of peaks experienced more pronounced perturbations upon MEEVD addition compared to p23 (Supplementary Fig. 6). Furthermore, additional resonances from

the TPR domain were perturbed upon p23 addition (Supplementary Fig. 6c, d).

### Structure of the p23-FKBP51 complex

To gain high-resolution insights into the structure of the p23-FKBP51 complex, we modeled the complex using HADDOCK docking[41,42]. The docking was performed with the crystal structures of p23 (1EJF)[16] and FKBP51 (5NJX)[40]. To calculate the structure of the complex, the residues of p23 that showed the highest CSPs (11, 56, 88, 94, 103, 107) were

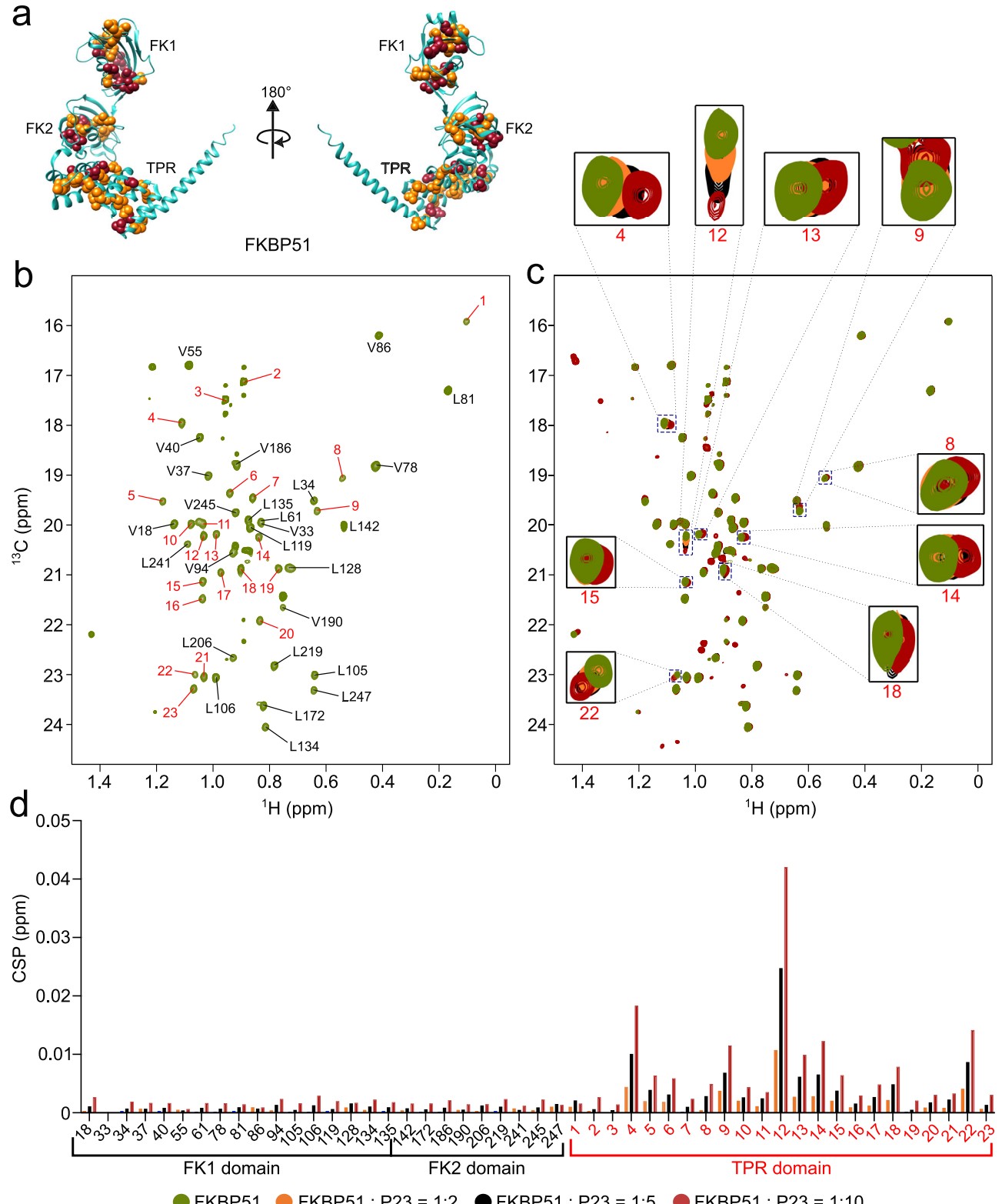

**Fig. 3 | p23 recognizes the TPR domain of FKBP51. a** Crystal structure (PDB: 5NJX[40]) of FKBP51. Leucine and valine residues are shown with yellow and red spheres, respectively. **b** 2D $^1$H-$^{13}$C HMQC spectra of the $^{13}$C labeled methyl groups of the leucine and valine residues of FKBP51. Assignments of the residues present in the FK1 and FK2 domains are indicated in the spectrum. The residues present in the TPR domain couldn't be assigned and are indicated by peak numbers from 1 to 23. **c** Perturbation of the cross-peaks of the methyl groups of FKBP51 upon addition of two-fold (orange), five-fold (black), and ten-fold (red) molar excess of unlabeled p23.

The new peaks that appeared in the spectra upon the addition of p23 represent peaks of p23 appearing at high concentrations due to the natural abundance of $^{13}$C. **d** Chemical shift perturbations (CSP) of the cross peaks of the methyl groups of FKBP51 (**c**) upon addition of two-fold (orange), five-fold (black), and ten-fold (red) molar excess of unlabeled p23. The CSPs of residues present in the TPR domain are indicated by peak numbers from 1 to 23 as shown in (**b**). Source data are provided as a Source Data file.

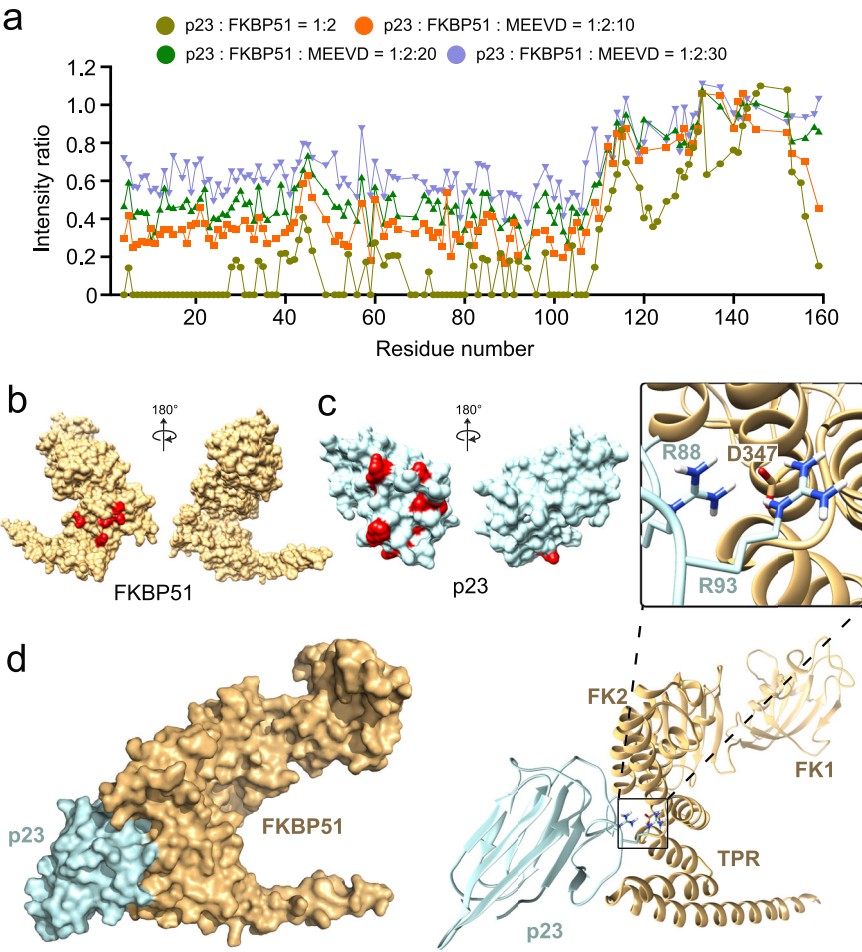

**Fig. 4 | Structure of the p23-FKBP51 complex. a** Changes in the intensities of the cross-peaks in the TROSY-HSQC spectra of [15]N-labeled p23 upon the addition of two-fold (yellowish green) molar excess of FKBP51. The intensities of the cross-peaks in the p23-FKBP51 complex gradually recover upon the addition of ten-fold (orange), twenty-fold (green), and thirty-fold (purple) molar excess of the C-terminal peptide (MEEVD) of Hsp90. **b** Crystal structure (PDB: 5NJX[40]) of FKBP51.

The binding site of the MEEVD peptide is highlighted in red. **c** Crystal structure (PDB: 1EJF[16]) of p23. The binding site of the FKBP51 peptide is highlighted in red. **d** Structural model of the p23-FKBP51 complex calculated by HADDOCK. R88 and R93 of p23 interact with D347 of FKBP51 in the complex structure. Source data are provided as a Source Data file.

assigned as the active site on p23 and the residues of FKBP51 that bind to the MEEVD peptide (K272, E273, K352, N322, K329) were assigned as the active site on FKBP51 (Fig. 4b, c). The analysis generated 92 structures within a single cluster with an RMSD deviation of 0.9 ± 0.5 Å from the lowest energy structure (Supplementary Table 1). The lowest energy structure comprises the C-terminal structured domain of p23 in contact with the MEEVD-binding site on the TPR domain of FKBP51 (Fig. 4d). Analysis of the binding interface revealed an insertion of the side chains of R88 and R93 of p23 into a binding pocket in the TPR domain of FKBP51, including a potential direct contact with D347 of FKBP51 (Fig. 4d). Consistent with the formation of this binding interface, A94 the residue next to R93 (which could not be assigned in the NMR spectra) and R88 of p23 showed the highest CSPs in the presence of FKBP51 (Fig. 2a).

To validate the structural model of the p23-FKBP51 complex, we mutated both the arginine residues involved in binding with FKBP51, i.e., R88 and R93 to alanine. To get insight into the binding of the R88A/R93A-mutant p23 with FKBP51, we added equimolar or two-fold molar excess of unlabeled FKBP51 to [15]N-labeled R88A/R93A-mutant p23. Unlike the case of wild-type p23, the addition of FKBP51 to the mutant p23 caused very little attenuation of the NMR signals from the folded domain of p23 indicative of a drastic reduction in the binding of the two proteins (Supplementary Fig. 7). Also, most of the signals near the C-terminal end of the folded domain of p23, the major binding site of

FKBP51 on p23, remained unperturbed (Supplementary Fig. 7b) further validating the structural model of the p23-FKBP51 complex.

## Contribution of the C-terminal tail of p23 to the p23-FKBP51 complex

To further characterize the p23-FKBP51 complex, we performed size-exclusion chromatography coupled with small angle X-ray scattering (SEC-SAXS) with both the free FKBP51 and the FKBP51-p23 complex. Analysis of the SAXS data of FKBP51 alone generated a density map characterized by a radius of gyration (Rg) of 39.5 Å and a maximum density (Dmax) of 139.4 Å (Fig. 5a–e). The crystal structure of FKBP51 fits well into the SAXS-derived density map (Fig. 5e, f). An additional density at the C-terminus of FKBP51 is observed which may be attributed to the C-terminal 32 residues (residues 426–457) of FKBP51 that are absent in the crystal structure (Fig. 5f).

The SAXS data of the p23-FKBP51 complex generated a density map with Rg 40.8 Å and Dmax 155.6 Å. Although an equimolar ratio of p23 and FKBP51 were injected into the SEC column, most of the p23 remained unbound (Fig. 5a). Thus, the scattering data for the FKBP51-p23 complex represents a mixture of species, with a minor contribution from the FKBP51-p23 complex to the overall FKBP51 scattering (Fig. 5g). The increase in Rg and Dmax is in agreement with the complex formation between FKBP51 and p23 (Fig. 5b–g). Further comparison of the density map of the free FKBP51 and p23-FKBP51 complex

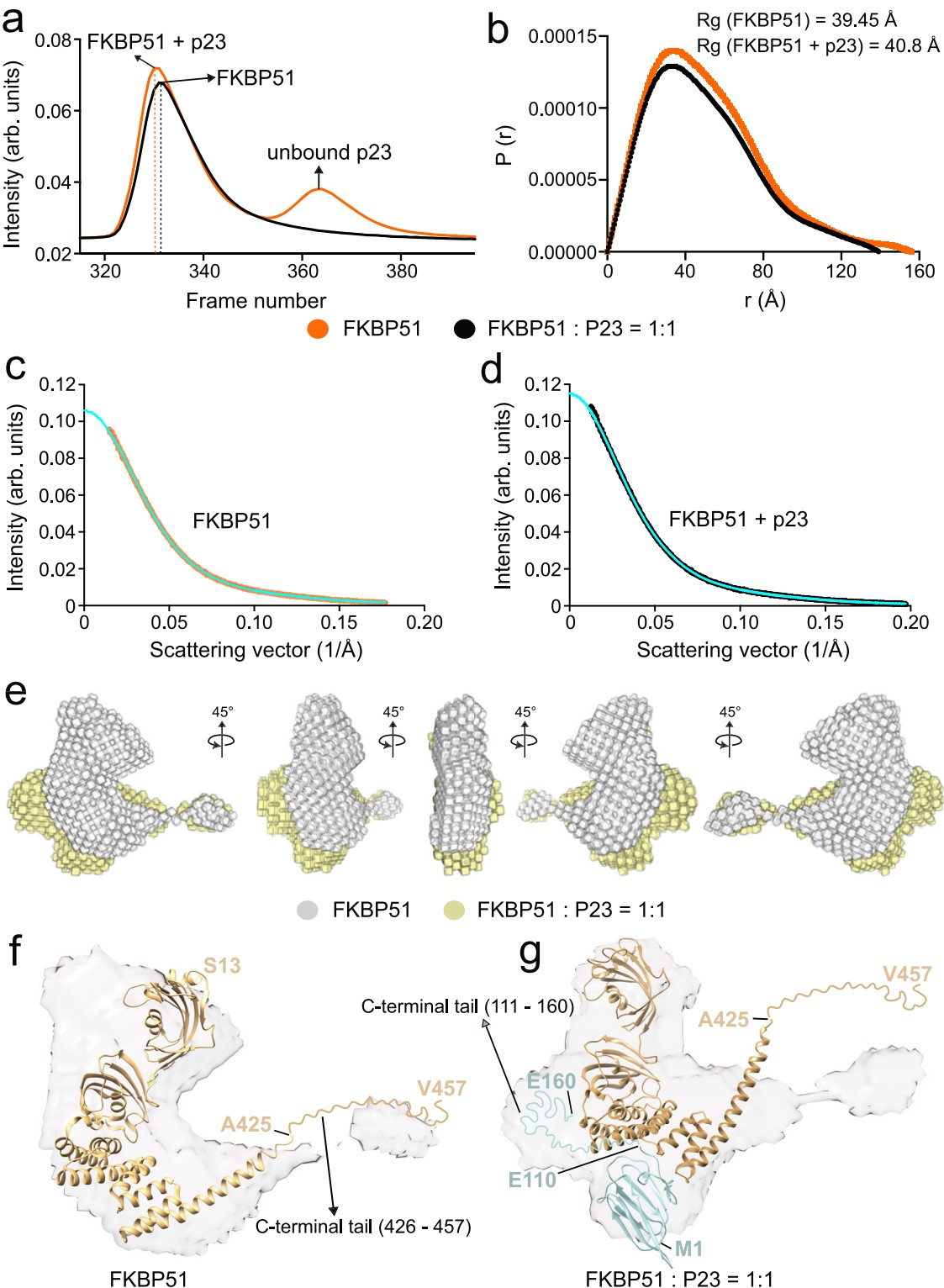

**Fig. 5 | Analysis of the p23-FKBP51 complex by SEC-SAXS. a** Elution profile of free FKBP51 (black) and the p23-FKBP51 complex (orange) from the size exclusion chromatography. **b** P(r) distribution curve of the free FKBP51 (black) and the p23-FKBP51 complex (orange) calculated from SAXS. The values of the radius of gyration (Rg) of free FKBP51 and the p23-FKBP51 complex are indicated. **c** Scattering intensity vs scattering vector plot for the SAXS data of FKBP51. The scattering data is shown in orange and the fit used to generate the 3D model of FKBP51 is shown in sky blue. **d** Scattering intensity vs scattering vector plot for the SAXS data of FKBP51-p23 complex. The scattering data is shown in black and the fit used to generate the 3D model of FKBP51-p23 complex is shown in sky blue.

**e** Superimposition of the SAXS-derived density maps of free FKBP51 (gray) and p23-FKBP51 complex (yellow). **f** Superimposition of the crystal structure of FKBP51 (PDB: 5NJX[40]) with the SAXS-derived density map of free FKBP51. The C-terminal residues 426-457 that are absent in the crystal structure of FKBP51 are represented as a disordered tail. **g** Superimposition of the p23-FKBP51 complex structure (Fig. 4d) with the density map of the p23-FKBP51 complex. The C-terminal residues that are absent in the crystal structures of free p23 and free FKBP51 are represented as disordered cyan (residues 111–160) or yellow (residues 426–457) tails, respectively. Source data are provided as a Source Data file.

revealed additional electron density near the TPR domain of FKBP51 in the complex state (Fig. 5e). The complex structure of p23-FKBP51 (Fig. 4d) also fits into the density map calculated from SAXS although the disordered tail of FKBP51, and potentially also the C-terminal helix of FKBP51, may undergo conformational changes (Fig. 5g).

Additionally, we observed density near the FK2 and TPR domain of FKBP51 for the p23-FKBP51 complex. This additional density may be attributed to the presence of the 50 residue-long C-terminal tail of p23 that is missing in the crystal structure of free p23 (Fig. 5g). Together with the contribution of the disordered C-terminal residues of p23 to p23-FKBP51 complex formation revealed by NMR spectroscopy (Fig. 2b, c), the data suggest binding of the C-terminal tail of p23 to the FK2/TPR interface region in the p23-FKBP51 complex.

### p23-FKBP51-tau trimeric complex formation

The findings from our study reveal a crucial role played by p23 in delaying aggregation of tau (Fig. 1b, c). To determine the molecular basis of the interaction between tau and p23, we first titrated $^{15}$N-labeled tau with two, five, and ten-fold molar excess of unlabeled p23 and observed decreased intensity as well as CSPs of the residues of tau present in the P2 domain as well as the repeat domain suggesting binding of p23 to an extended interface of tau (Fig. 6a). The strongest CSPs were detected in the proline-rich region P2 that is the most positively charged region of tau. In contrast, the addition of five-fold and ten-fold molar excess of the truncated (1–119) mutant p23, which lacks the C-terminal disordered tail, did not lead to any changes in the intensity or chemical shifts of tau signals (Fig. 6b). The C-terminal disordered tail of p23 thus interacts with the predominantly positively charged C-terminal half of tau.

To further confirm the interaction of the C-terminal tail of p23 with tau, we added five-fold and ten-fold molar excess of unlabeled tau to $^{15}$N-labeled p23 and observed a global decrease in intensity of the p23 signals in agreement with complex formation. In addition to the intensity decrease, we also observed signal perturbations of residues ~130–160 at the C-terminal end of the disordered tail of p23 supporting a direct binding of tau to these residues (Fig. 6c). We also performed another experiment with $^{15}$N-labeled (1–119)-mutant p23 in the presence of five-fold and ten-fold molar excess of unlabeled tau (Fig. 6d). The NMR signals of the C-terminally truncated p23 did not change upon addition of tau, indicating that no interaction occurred. The combined data demonstrate that tau interacts with the C-terminal disordered tail of p23.

Our data reveal that FKBP51 binds near the C-terminal end of the folded domain of p23, potentially also engaging the α-helical structure formed by residues (114–130), while tau binds to the more C-terminal residues (~130–160) of p23. Thus, all three proteins may form a trimeric complex of FKBP51-p23-tau (Fig. 6e). In the trimeric complex, the positively charged repeat-domain of tau can interact with the predominantly negatively charged disordered residues (~130–160) of p23. As previously reported[32], tau's proline-rich domain can cluster around the catalytically active FK1 domain of FKBP51, which possesses PPIase activity, thereby further contributing to a trimeric interaction.

To provide experimental validation for the formation of the FKBP51-p23-tau complex, we combined the three proteins at an equimolar ratio and stabilized the resulting trimeric complex by introducing the chemical crosslinker disuccinimidyl suberate (DSS). Upon loading the crosslinked complex onto an SDS-PAGE gel, a distinct band above 220 kDa emerged (Supplementary Fig. 8a). Subsequently, this band was excised and subjected to Mass Spectrometry analysis. The analysis unveiled a network of intermolecular crosslinks among p23, FKBP51, and tau, conclusively confirming the formation of the trimeric complex (Supplementary Fig. 8, Supplementary Data 1). The crosslinks observed between p23 and FKBP51 further validate the structural model of p23-FKBP51 complex (Supplementary Fig. 9). Apart from the band of the trimeric FKBP51-p23-tau complex, several other bands

appear in the SDS-PAGE gel suggesting the presence of heterogenous complexes potentially between FKBP51-tau, tau-p23, and p23-FKBP51 (Supplementary Fig. 8a, Supplementary Data 1).

## Discussion

The identification and understanding of the molecular mechanisms of chaperones that modulate the aggregation of tau may provide an entry point to develop novel therapeutic approaches. However, tau is an engrossing target for the Hsp90 chaperone machinery as it is unfolded in its native state thereby challenging the paradigm of protein folding and maturation by chaperones. Although the role played by Hsp90 in the progression of tauopathies is complex, the current understanding indicates that Hsp90 drives tau accumulation and neurotoxicity[43]. One of the hypotheses for the Hsp90-mediated toxicity is that by attempting to chaperone the microtubule-binding domain of tau, Hsp90 increases the half-life of tau thereby increasing its chances to form toxic aggregates[44]. Also, the interaction of Hsp90 with tau is reported to expose the aggregation-prone repeat domain of tau leading to the formation of toxic oligomers[45]. The inhibition of Hsp90 activity by geldanamycin induces the activation of HSF-1, which further induces the expression of Hsp70 and Hsp40 leading to the reduction of aggregated proteins[46,47]. However, inhibition of Hsp90 may be associated with adverse effects as Hsp90 interacts with 60% of the protein kinases and plays a crucial role in cellular stress response[46]. Thus, finding more specific targets, potentially co-chaperones of Hsp90 that selectively interact with tau or other IDPs, might help to circumvent some of these problems.

Using our recently developed co-factor-free aggregation assay of tau, we showed that the Hsp90 co-chaperone p23 strongly delays tau's aggregation while FKBP51, another Hsp90 co-chaperone, did not slow tau's amyloid formation kinetics (Fig. 1). While further studies are required, a tug-of-war between two different activities of FKBP51 may be present, the PPIase activity of the FK1 domain and the chaperone activity of FKBP51's TPR domain. We speculate that the PPIase activity of FKBP51 may enhance the aggregation of tau whereas the chaperone activity may delay tau's aggregation. Such opposing roles on the aggregation propensity of tau and α-synuclein were reported for the two PPIases FKBP12 and PPIA/CypA, respectively[48,49]. The two activities generally depend on the concentration of the chaperone/co-chaperone, with a lower concentration favoring PPIase activity while at higher concentrations the chaperone activity may dominate[48,49]. Thus, at the concentration of FKBP51 used in this study, the PPIase and chaperone activities might counterbalance each other, resulting in no significant effect on tau aggregation kinetics. Currently, we cannot further investigate this hypothesis because of the lower stability of FKBP51 at higher concentrations in our aggregation assay. Notably, FKBP51 stimulates tau aggregation in vivo[33], suggesting that its PPIase activity may play an important role in vivo.

In contrast to FKBP51, p23 slows the fibrillization kinetics of tau even at substoichiometric concentrations (Fig. 1). p23's aggregation inhibition activity can be attributed to the interaction of the negatively charged C-terminal tail of p23 with tau's positively charged aggregation-prone repeat domain (Fig. 6a–d). The C-terminal tail of p23 has previously been suggested to exert chaperone activity[26]. Our results further revealed that the presence of both p23 and FKBP51 attenuated tau's aggregation to a slightly larger extent than p23 alone at comparable concentrations (Fig. 1). This may arise from the formation of a trimeric complex between tau, p23, and FKBP51 in which tau's repeat-domain interacts with the C-terminal tail of p23 and tau's proline-rich domain clusters at the FK1 domain of FKBP51 (Fig. 6e, Supplementary Fig. 8). The trimeric p23-FKBP51-tau interaction is most likely not a single rigid complex, but rather a dynamic ensemble of trimeric complex structures in which different positively charged regions of Tau bind to the negatively charged C-terminal tail of p23.

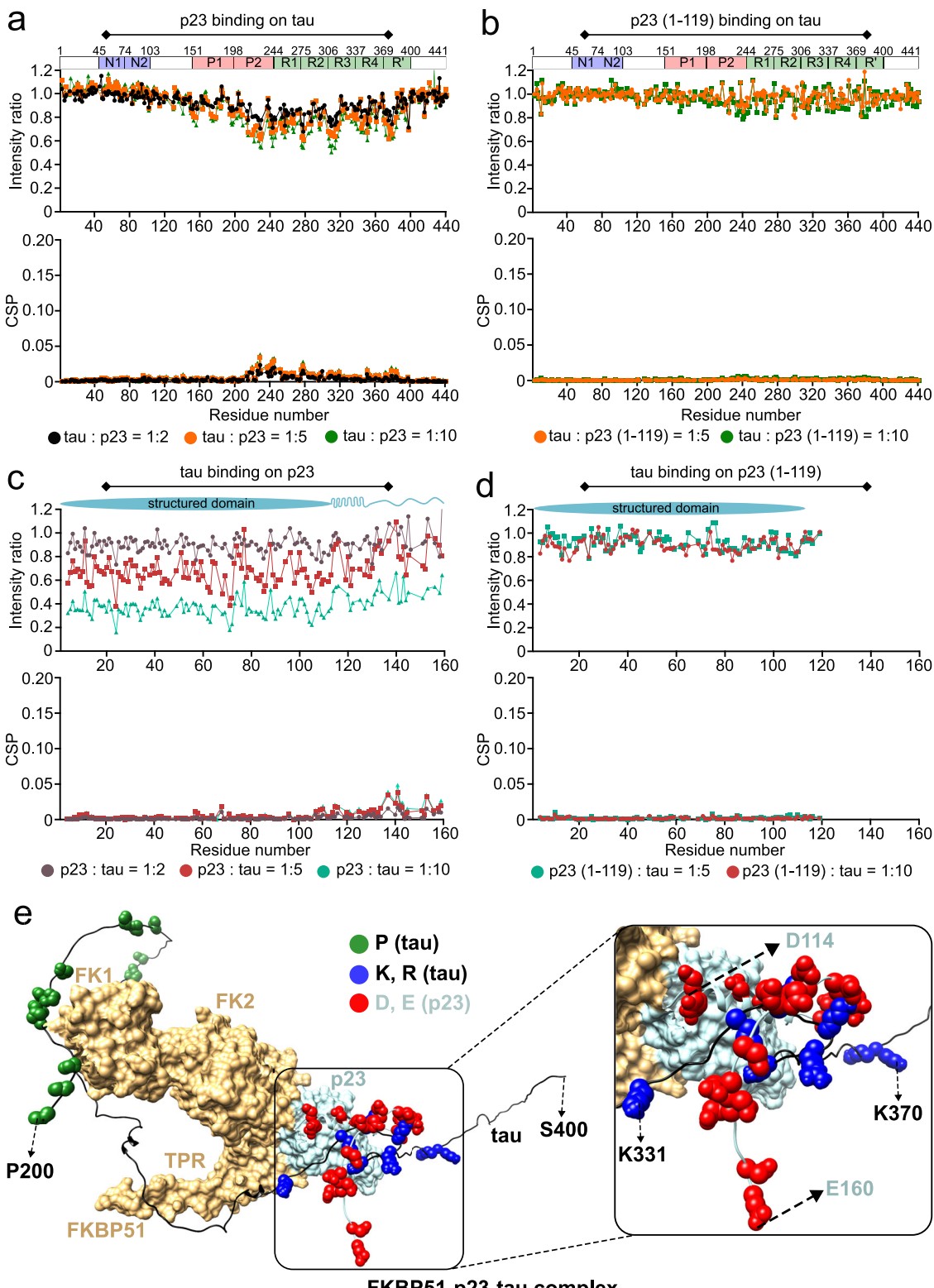

**FKBP51-p23-tau complex**

We determined the structural basis of the p23-FKBP51 complex using an integrative structural biology approach. The reliability of the modeled complex structure was supported by site-directed mutagenesis of two arginine residues of p23 that insert into a binding pocket on FKBP51 (Figs. 4 and 5). We further showed that p23 and the C-terminal peptide of Hsp90 (MEEVD) bind to an overlapping site on FKBP51's TPR domain (Fig. 4a). As Hsp90's MEEVD residues play a crucial role in mediating the protein's interaction with FKBP51[32,50], p23 and Hsp90 may potentially compete to bind to FKBP51. Concurrently,

FKBP51 is reported to synergize with Hsp90 to promote toxic tau oligomerization by forming an Hsp90-FKBP51-tau pro-toxic complex[32,33]. The formation of a protective p23-FKBP51-tau trimeric complex may thus be a regulation mechanism to counteract toxic Hsp90-FKBP51 mediated toxicity (Fig. 7). While p23 alone can exert a protective role in regulating tau aggregation (Figs. 1 and 7), the p23-FKBP51 complex can potentially counteract the FKBP51-mediated tau toxicity.

Taken together, our findings reveal an Hsp90-independent, p23-FKBP51-mediated chaperoning of tau protein.

**Fig. 6 | Formation of p23-FKBP51-tau trimeric complex. a** Changes in the intensities (top) and chemical shift perturbations (CSPs) (bottom) of the cross peaks in the $^1$H-$^{15}$N HSQC spectrum of $^{15}$N-labeled tau upon the addition of two-fold (black), five-fold (orange), and a ten-fold (green) molar excess of unlabeled p23. The domain diagram of tau is shown at the top. **b** Changes in the intensities (top) and chemical shift perturbations (CSPs) (bottom) of the cross peaks in the $^1$H-$^{15}$N HSQC spectrum of $^{15}$N-labeled tau upon the addition of five-fold (orange), and a ten-fold (green) molar excess of unlabeled p23 (1-119). The domain diagram of tau is shown at the top. **c** Changes in the intensities (top) and chemical shift perturbations (CSPs) (bottom) of the cross peaks in the TROSY $^1$H-$^{15}$N HSQC spectrum of deuterated, $^{15}$N-labeled p23 upon the addition of two-fold (brown), five-fold (red), and a ten-fold (bluish-green) molar excess of unlabeled tau. The domain diagram of

p23 is shown at the top. **d** Changes in the intensities (top) and chemical shift perturbations (CSPs) (bottom) of the cross peaks in the TROSY $^1$H-$^{15}$N HSQC spectrum of deuterated, $^{15}$N-labeled p23 (1-119) upon the addition of five-fold (bluish-green), and a ten-fold (red) molar excess of unlabeled tau. The domain diagram of the folded domain of p23 is shown at the top. **e** A conceptual model of the p23-FKBP51-tau trimeric complex. The proline-rich domain of tau interacts with the catalytically active FK1 domain of FKBP51 and tau's repeat domain interacts with the C-terminal tail of p23. The proline residues of tau are shown with green spheres, the lysine and arginine residues of tau are shown with blue spheres, and the aspartate and glutamate residues of p23 are shown with red spheres. Source data are provided as a Source Data file.

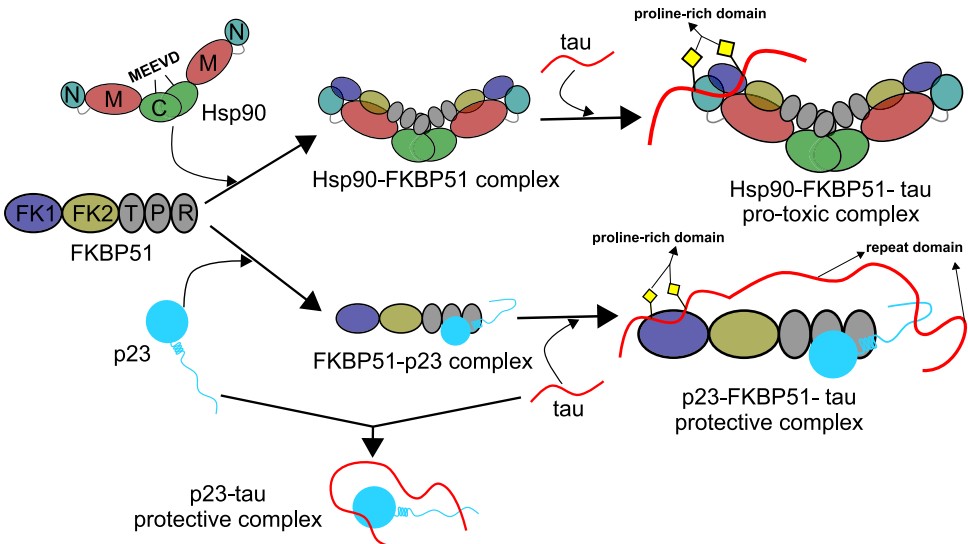

**Fig. 7 | Competition of p23 and Hsp90 for FKBP51 modulates interactions with tau.** The MEEVD residues present at the C-terminus of Hsp90 mediate its binding with FKBP51 thereby forming the Hsp90-FKBP51 complex. The Hsp90-FKBP51 complex can further bind to the client tau and form a Hsp90-FKBP51-tau pro-toxic complex. On the other hand, p23 can compete with Hsp90 and bind to the TPR domain of FKBP51. This p23-FKBP51 complex can further bind to tau and form a p23-FKBP51-tau trimeric complex potentially exerting a protective role against Hsp90-FKBP51 mediated toxicity. The protective role of p23 alone by directly interacting with tau is also shown. Of note, binary p23-Hsp90 and ternary p23-FKBP51-Hsp90 complexes also exist (not shown).

## Methods

### Protein purification

Mutants of p23 were obtained by site-directed mutagenesis using a thermocycler (SensoQuest Labcycler). Phusion high-fidelity PCR master mix (Thermofisher) was used to perform PCR reactions. The sequence of the primers used to generate the mutants are available in Supplementary Table 2.

To prepare unlabeled wild-type (WT) p23, the truncated mutant p23 (1–119), as well as the R88A/R93A-mutant p23 a single colony from the LB-agar plate was taken and grown overnight in 50 mL LB medium supplemented with 30 μg/mL kanamycin at 37 °C. 10 mL of the overnight culture was transferred to 1 L LB medium supplemented with 30 μg/mL kanamycin and allowed to grow at 37 °C until an OD$_{600}$ of 0.8–0.9 was reached. Subsequently, the cells were induced with 0.1 mM IPTG and expressed overnight at 20 °C.

To obtain uniformly $^{15}$N-labeled/ deuterated-$^{15}$N-labeled p23 WT/ p23 (1–119)/ p23 R88A R93A, cells were grown in 1 L M9 medium prepared in H$_2$O/D$_2$O and supplemented with 1 g/L $^{15}$NH$_4$Cl as the only nitrogen source and 4 g/L $^{12}$C-glucose/ D-Glucose-1,2,3,4,5,6,6-d$_7$ at 37 °C until an OD$_{600}$ of 0.8–0.9 was reached. Subsequently, the cells were induced with 0.1 mM IPTG and expressed overnight at 20 °C.

After harvesting, the cell pellets were resuspended in lysis buffer (20 mM Tris, pH 8.0, 500 mM NaCl, 10 mM imidazole, 3 mM βME) complemented with protease inhibitor mixture and 1 mM PMSF. Subsequently, cells were disrupted by sonication (6 min, 30% amplitude, pulse on - 10 s, pulse off – 15 s), and the cell debris was

pelleted by centrifugation at 58500 g at 4 °C for 30 min. The supernatant was then loaded into a His-trap column and the bound protein was eluted with elution buffer (20 mM Tris, pH 8.0, 500 mM NaCl, 500 mM imidazole, 3 mM βME). The protein was dialyzed against 20 mM Tris, pH 8.0, 100 mM NaCl, 6 mM βME, and 37 NIH thrombin was added per mg of protein to cut the His-tag. The protein-thrombin mixture was incubated for 1 h at room temperature and then the solution was again passed through the His-trap column to remove any uncut protein. Finally, the His-tag cut protein was purified by size exclusion chromatography using HiLoad -SD 75, 16/600 column in 20 mM Tris, pH 7.0, 100 mM NaCl, and 1 mM DTT buffer.

To prepare unlabeled FKBP51/ FK1-FK2, a single colony from the LB-agar plate was taken and grown overnight in 50 mL LB medium supplemented with 30 μg/mL kanamycin at 37 °C. 10 mL of the overnight culture was transferred to 1 L LB medium supplemented with 30 μg/mL kanamycin and allowed to grow at 37 °C until an OD$_{600}$ of 0.8–0.9 was reached. Subsequently, the cells were induced with 0.3 mM IPTG and expressed overnight at 25 °C.

To prepare LV-labeled FKBP51, cells were grown in 1 L M9 medium prepared in D$_2$O and supplemented with 1 g/L $^{14}$NH$_4$Cl and 4 g/L D-Glucose-1,2,3,4,5,6,6-d$_7$ at 37 °C until an OD$_{600}$ of 0.6 was reached. Then, DLAM LV pro-S (NMR-Bio) was added as a source to label the methyl groups of Leucine and Valine residue by $^{13}$C isotope and after one hour the cells were induced with 0.3 mM IPTG and expressed overnight at 25 °C.

After harvesting, the cell pellets were resuspended in lysis buffer (20 mM Tris, pH 8.0, 500 mM NaCl, 10 mM imidazole, 6 mM βME) complemented with protease inhibitor mixture and 1 mM PMSF. Subsequently, cells were disrupted by sonication (5 min, 30% amplitude, pulse on - 2 s, pulse off – 6 s), and the cell debris was pelleted by centrifugation at 40,000 g at 4 °C for 30 min. The supernatant was then loaded into a His-trap column and the bound protein was eluted with elution buffer (20 mM Tris, pH 8.0, 500 mM NaCl, 250 mM imidazole, 6 mM βME). To cut the His-tag HRV 3 C protease (Thermo-Fisher Scientific) was added to a molar ratio of 1:800 (enzyme: protein) and the mixture was dialyzed against 20 mM Tris, pH 7.5, 150 mM NaCl, 0.5 mM EDTA, 1 mM DTT buffer. To remove the uncut protein the solution was again passed through the His-trap column. Finally, the His-tag cut protein was purified by size exclusion chromatography using HiLoad -SD 75, 26/600 column in 50 mM NaP, pH 6.4, 500 mM NaCl, and 10 mM DTT buffer.

To prepare unlabeled 2N4R tau protein, a single colony from the LB-agar plate was taken and grown overnight in 50 mL LB medium supplemented with 100 µg/mL ampicillin at 37 °C. 22 mL of the overnight culture was transferred to 1 L LB medium supplemented with 100 µg/mL ampicillin and allowed to grow until an $OD_{600}$ of 0.8–0.9 was reached. Subsequently, the cells were induced with 0.5 mM IPTG and expressed for 1 h.

To obtain uniformly $^{15}$N-labeled 2N4R tau, cells were grown in 8 L LB until an $OD_{600}$ of 0.6–0.8 was reached, then centrifuged at low speed (5,000 g), washed with 1× M9 salts, and resuspended in 2 L M9 minimal medium supplemented with 1 g/L $^{15}NH_4Cl$ as the only nitrogen source. After 1 h, the cells were induced with 0.5 mM IPTG and expressed overnight at 37 °C.

After harvesting, cell pellets were resuspended in lysis buffer (20 mM MES (pH 6.8), 1 mM EGTA, 2 mM DTT) complemented with protease inhibitor mixture, 0.2 mM $MgCl_2$, lysozyme, and DNAse I. Subsequently, cells were disrupted with a French pressure cell press (in ice-cold conditions to avoid protein degradation). NaCl was added to a final concentration of 500 mM, and lysates were boiled for 20 min. Denatured proteins were removed by ultracentrifugation with 127,000 g at 4 °C for 30 min. To precipitate the DNA, 20 mg/mL streptomycin sulfate was added to the supernatant and incubated for 15 min at 4 °C followed by centrifugation at 15,000 g for 30 min. The pellet was discarded, and tau protein was precipitated by adding 0.361 g/mL ammonium sulfate to the supernatant, followed by centrifugation at 15,000 g for 30 min. The pellet containing tau protein was resuspended in buffer A (20 mM MES (pH 6.8), 1 mM EDTA, 2 mM DTT, 0.1 mM PMSF, 50 mM NaCl) and dialyzed against the same buffer (buffer A) to remove excess salt. The next day, the sample was filtered and applied to an equilibrated ion-exchange chromatography column (Mono S 10/100 GL, GE Healthcare), and weakly bound proteins were washed out with buffer A. Tau protein was eluted with a linear gradient of 60% final concentration of buffer B (20 mM MES pH 6.8, 1 M NaCl, 1 mM EDTA, 2 mM DTT, 0.1 mM PMSF). Protein samples were concentrated by ultrafiltration (5 kDa Vivaspin, Sartorius) and further purified by reverse phase chromatography using a preparative C4 column (Vydac 214 TP, 5 µm, 8 ×250 mm) in an HPLC system coupled with ESI mass spectrometer. Protein purity was confirmed using mass spectrometry, and the purified protein was lyophilized and redissolved in the buffer of interest.

## Aggregation assay

The aggregation of 2N4R tau in the absence or presence of co-chaperones was performed using a previously described co-factor-free aggregation protocol[34]. Briefly, 25 µM of 2N4R tau in the absence or presence of co-chaperones were aggregated at 37 °C in 25 mM HEPES, 10 mM KCl, 5 mM $MgCl_2$, 3 mM TCEP, 0.01% $NaN_3$, pH 7.2 buffer in a 96 well plate using a Tecan spark plate reader. Three PTFE beads along with double orbital shaking were used to promote fibrillization.

Thioflavin-T (ThT) at a final concentration of 50 µM was used to monitor the aggregation kinetics.

## Fluorescence dye binding

30 µM of the fluorescent dye curcumin was added to 5 µM preformed tau fibrils (aggregated either in the absence or presence of different co-chaperones). The samples were transferred to a 96-well plate and the fluorescent measurements were performed on a Tecan spark plate reader. The excitation frequency was set to 440 nm and the emission spectra were recorded from 485 nm to 800 nm at a step size of 2 nm.

## NMR Spectroscopy

$^1$H-$^{15}$N TROSY HSQC spectra of deuterated, uniformly $^{15}$N-labeled p23 WT (40 µM) in the absence or presence of equimolar, two-fold, three-fold molar excess of unlabeled FKBP51 were recorded in 25 mM HEPES, 10 mM KCl, 5 mM $MgCl_2$, 1 mM DTT, pH 7.2 buffer at 303 K on an Avance Neo 700 MHz spectrometer (Bruker) using a 5 mm TCI (H/C/N) Cryoprobe. The spectra were collected with 192 scans per point (ns), and acquisition times td1 = 56.3 ms and td2 = 112.6 ms.

$^1$H-$^{15}$N TROSY HSQC spectra of deuterated, uniformly $^{15}$N-labeled p23 (1–119) (25 µM) in the absence or presence of equimolar, two-fold, five-fold molar excess of unlabeled FKBP51 were recorded in 25 mM HEPES, 10 mM KCl, 5 mM $MgCl_2$, 1 mM DTT, pH 7.2 buffer at 303 K on an Avance Neo 800 MHz spectrometer (Bruker) using a 5 mm TCI (H/C/N) Cryoprobe. The spectra were collected with 128 scans per point (ns), and acquisition times td1 = 24.2 ms and td2 = 49.1 ms.

$^1$H-$^{13}$C HMQC/methyl-TROSY spectra of $^{13}$C methyl (LV) labeled FKBP51 (50 µM) in the absence or presence of two-fold, five-fold, ten-fold molar excess of unlabeled p23 or five-fold, fifteen-fold molar excess of MEEVD peptide were recorded in 25 mM HEPES, 10 mM KCl, 5 mM $MgCl_2$, 1 mM DTT, pH 7.6 (in 100% $D_2O$) buffer at 308 K on an Avance III 900 MHz spectrometer (Bruker) using a 5 mm TCI (H/C/N) Cryoprobe. The spectra were collected with 128 scans per point (ns), and acquisition times td1 = 56.5 ms and td2 = 87.3 ms.

$^1$H-$^{15}$N TROSY HSQC spectra of a mixture of uniformly $^{15}$N-labeled p23 WT (30 µM) and unlabeled FKBP51 (60 µM) in the absence or presence of ten-fold, twenty-fold, thirty-fold molar excess of unlabeled MEEVD peptide were recorded in 25 mM HEPES, 10 mM KCl, 5 mM $MgCl_2$, 1 mM DTT, pH 7.2 buffer at 303 K on an Avance III 600 MHz spectrometer (Bruker) using a 5 mm QCI (H/C/N/F) Cryoprobe. The spectra were collected with 200 scans per point (ns), and acquisition times td1 = 60.1 ms and td2 = 94.2 ms.

$^1$H-$^{15}$N TROSY HSQC spectra of deuterated, uniformly $^{15}$N-labeled p23 R88A R93A (40 µM) in the absence or presence of equimolar, and two-fold molar excess of unlabeled FKBP51 were recorded in 25 mM HEPES, 10 mM KCl, 5 mM $MgCl_2$, 1 mM DTT, pH 7.2 buffer at 303 K on an Avance Neo 800 MHz spectrometer (Bruker) using a 5 mm TCI (H/C/N) Cryoprobe. The spectra were collected with 176 scans per point (ns), and acquisition times td1 = 24.2 ms and td2 = 49.1 ms.

$^1$H-$^{15}$N HSQC spectra of uniformly $^{15}$N-labeled tau (20 µM) either in the absence or presence of five-fold, ten-fold molar excess of unlabeled p23 were recorded in 50 mM NaP, 10 mM NaCl, 1 mM DTT, pH 6.8 buffer at 278 K on an Avance III 900 MHz spectrometer (Bruker) using a 5 mm TCI (H/C/N) Cryoprobe. The spectra were collected with 40 scans per point (ns), and acquisition times td1 = 110.3 ms and td2 = 94.6 ms.

$^1$H-$^{15}$N HSQC spectra of uniformly $^{15}$N-labeled tau (30 µM) either in the absence or presence of five-fold, ten-fold molar excess of unlabeled p23 (1–119) were recorded in 50 mM NaP, 10 mM NaCl, 1 mM DTT, pH 6.8 buffer at 278 K on an Avance Neo 700 MHz spectrometer (Bruker) using a 5 mm TCI (H/C/N) Cryoprobe. The spectra were collected with 40 scans per point (ns), and acquisition times td1 = 150.2 ms and td2 = 133.1 ms.

$^1$H-$^{15}$N TROSY HSQC spectra of deuterated, uniformly $^{15}$N-labeled p23 WT (30 µM) in the absence or presence of two-fold, five-fold, ten-

 

fold molar excess of unlabeled tau were recorded in 25 mM HEPES, 10 mM KCl, 5 mM MgCl$_2$, 1 mM DTT, pH 7.2 buffer at 303 K on an Avance Neo 700 MHz spectrometer (Bruker) using a 5 mm TCI (H/C/N) Cryoprobe. The spectra were collected with 224 scans per point (ns), and acquisition times td1 = 56.3 ms and td2 = 112.6 ms.

$^{1}$H-$^{15}$N TROSY HSQC spectra of deuterated, uniformly $^{15}$N-labeled p23 (1–119) (25 μM) in the absence or presence of five-fold, ten-fold molar excess of unlabeled tau were recorded in 25 mM HEPES, 10 mM KCl, 5 mM MgCl$_2$, 1 mM DTT, pH 7.2 buffer at 303 K on an Avance Neo 800 MHz spectrometer (Bruker) using a 5 mm TCI (H/C/N) Cryoprobe. The spectra were collected with 128 scans per point (ns), and acquisition times td1 = 24.2 ms and td2 = 49.1 ms.

Chemical shift assignments of p23[37], FKBP51[32,51], and tau[52] were previously reported. The spectra were recorded using Topsin 3.6.2/4.0.3 software (Bruker) and analyzed with CCPNMR 2.4.2 software[53].

Residue-specific intensity ratios were calculated according to an intensity ratio = (I/I$_0$), where I is the intensity of cross-peaks in the complex form and I$_0$ is the intensity of the cross-peaks of the free protein.

Chemical shift perturbations (CSP) were calculated according to Eq. 1:

$$CSP = \sqrt{0.5*\left\{(\partial H)^2 + (\partial N/5)^2\right\}} \tag{1}$$

## Isothermal titration calorimetry

Isothermal titration calorimetry (ITC) experiments were conducted using a Microcal PEAQ-ITC automated system (Malvern) at a constant temperature of 25 °C in a buffer comprising 25 mM HEPES, 10 mM KCl, 5 mM MgCl2, and 1 mM DTT at pH 7.2. The sample cell was loaded with 50 μM FKBP51 and titrated with 750 μM of p23. Each titration step involved a 0.4 μl initial injection followed by 18 injections of 2 μl each, with a stirring speed of 750 rpm and intervals of 150 s between injections. To account for the heat of dilution, a control experiment was performed by adding 750 μM of p23 to the buffer alone. Data analysis was performed using the Microcal PEAQ-ITC Analysis software, with thermodynamic parameters obtained by fitting a macroscopic binding model allowing for one set of binding sites.

## SAXS

SEC-SAXS experiments were performed with 150 μM of FKBP51 either in the absence or presence of equimolar p23. To perform size exclusion chromatography, samples were passed through a Shodex KW403-4F column. Analysis of the SEC-SAXS data was performed using CHROMIX software[54]. The shape of the proteins was determined using DAMMIF v1.1.2[55].

## Molecular docking

To gain high-resolution insights into the structure of the p23-FKBP51 complex, we modeled the complex using HADDOCK docking[41,42]. The docking experiment was performed using the HADDOCK webserver. The docking was performed with the crystal structures of p23 (1EJF)[16] and FKBP51 (5NJX)[40]. To calculate the structure of the complex, the residues of p23 that showed the highest CSPs (11, 56, 88, 94, 103, 107) were assigned as the active site on p23 and the residues of FKBP51 that bind to the MEEVD peptide (K272, E273, K352, N322, K329) were assigned as the active site on FKBP51.

## Chemical crosslinking

The complex was assembled by the addition of 1:1:1 (25 μM each) molar ratio of p23:FKBP51:tau protein. To crosslink the proteins, disuccinimidyl suberate (DSS, ThermoFischer Scientific; stock solubilized in DMSO) at a final concentration of 1 mM was added.

The protein-DSS mixture was incubated for 30 min at 25 °C and the rxn was stopped by the addition of 20 mM Tris, pH 7.4. 10 uL of the cross-linked complex was directly loaded into the SDS-PAGE gel. The respective band was carefully cut and kept in an Eppendorf tube. To wash the gel pieces, 150 μL of water was added and incubated for 5 min at 26 °C with 1050 rpm shaking in a thermomixer. The gel pieces were spun down and the liquid was removed using thin tips (the same washing protocol was used in all subsequent steps with different solvents). The gel pieces were washed again with 150 μL acetonitrile. After washing, the gel pieces were dried for 5 min using a SpeedVacc vacuum centrifuge. To reduce disulfide bridges, 100 μL of 10 mM DTT was added to the gel pieces and incubated for 50 min at 56 °C followed by centrifugation and removal of liquid. The gel pieces were washed again with 150 μL of acetonitrile. To alkylate reduced cysteine residues, 100 μL of 55 mM iodoacetamide were added and incubated for 20 min at 26 °C with 1050 rpm shaking followed by centrifugation and removal of liquid. Subsequently, the gel pieces were washed with 150 μL of 100 mM NH$_4$HCO$_3$, and then twice with 150 μL of acetonitrile and dried for 10 min in a vacuum centrifuge. The gel pieces were rehydrated at 4 °C for 45 min by addition of small amounts (2–5 μL) of digestion buffer 1 (12.5 μg/mL trypsin, 42 mM NH$_4$HCO$_3$, 4 mM CaCl$_2$). The samples were checked after every 15 min and more buffer was added in case the liquid was completely absorbed by the gel pieces. 20 μL of digestion buffer 2 (42 mM NH$_4$HCO$_3$, 4 mM CaCl$_2$) were added to cover the gel pieces and incubated overnight at 37 °C.

To extract the peptides, 15 μl water was added to the digest and incubated for 15 min at 37 °C with 1050 rpm shaking followed by spinning down the gel pieces. 50 μl acetonitrile was added to the entire mixture and incubated for 15 min at 37 °C with 1050 rpm shaking. The gel pieces were spun down and the supernatant (SN1) containing the extracted peptides was collected. 30 μl of 5% (v/v) formic acid was added to the gel pieces and incubated for 15 min at 37 °C with 1050 rpm shaking followed by spinning down. Again 50 μl acetonitrile were added to the entire mixture and incubated for 15 min at 37 °C with 1050 rpm shaking. The gel pieces were spun down and the supernatant (SN2) containing the extracted peptides was collected. Both supernatants (SN1 & SN2) containing the extracted peptides were pooled together and evaporated in the SpeedVacc vacuum centrifuge. The dried peptides were resuspended in 5% acetonitrile and 0.1% formic acid and analyzed using an Orbitrap Fusion Tribrid (Thermo Fischer Scientific) instrument.

For identification of crosslinked peptides, raw files were analyzed by pLink (v. 2.3.9)[56], using DSS as crosslinker and trypsin as digestion enzyme with maximal three missed cleavage sites. The search was conducted against a customized protein database containing all proteins within the complex. Carbamidomethylation of cysteines was set as a fixed modification, oxidation of methionines and acetylation at protein N-termini were set as a variable modifications. Searches were conducted in combinatorial mode with a precursor mass tolerance of 10 p.p.m. and a fragment ion mass tolerance of 20 p.p.m. The false discovery rate (FDR) was set to 0.01. Crosslinked sites were plotted as networks with xiNET[57].

## Reporting summary

Further information on research design is available in the Nature Portfolio Reporting Summary linked to this article.

## Data availability

The structures cited in this paper (2IOQ, 7L7I, 5NJX, 1EJF) are available from the protein data bank. NMR assignments referenced in this study can be accessed using the following accession codes: BMRB 6973, BMRB 6974, BMRB 50701, BMRB 19787. Source data are provided with this paper.

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

## Acknowledgements

We thank the mass spectrometry facility of Max Planck Institute for Multidisciplinary Sciences (MPINAT) Göttingen for the mass spectrometry data. SEC-SAXS experiments were performed at B21, Diamond Light Source, UK, and were supported by iNEXT, PID – 5962. M.Z. was supported by the European Research Council (ERC) under the EU Horizon 2020 research and innovation programme (grant agreement No. 787679).

## Author contributions

P.C. performed protein purification, site-directed mutagenesis, aggregation assays, NMR experiments, ITC experiments, crosslinking experiments, molecular docking, prepared samples for the SAXS experiment, and analyzed the SAXS data; M.Z. supervised the project; P.C. and M.Z. designed the project and wrote the paper.

## Funding

## Competing interests

The authors declare no competing interests.
