## [Transparent Peer Review file · Nature Communications]

Interplay of p23 FKBP51 and their chaperone complex in regulating tau aggregation

Corresponding Author: Professor Markus Zweckstetter

Version 0:

Reviewer comments:

Reviewer #1

(Remarks to the Author)

This article describes the interaction between the protein tau and two co-chaperone p23 and FKBP51. It mostly focuses on structural aspect of this interaction by identifying interaction sites. The report brings new data confirming direct interactions but does not provide breakthrough concepts or approaches to the field. It rather provides finer details on aggregation-modulating interactions with HSP90/FKBP51 that were already revealed, including by the same group a few years before (e.g.. Oroz et al. Nat Comm 2018).

The report is overall of good quality with some revision required as detailed below. One of the main conceptual concern is about the support for the notion of cooperativity as both binding and aggregation modulation occur without the need for the three partners. More details are provided below.

- About the effect of chaperones on aggregation

The authors stated that "The presence of both co-chaperones attenuated the aggregation of tau to a greater extent than the individual co-chaperones at the comparable concentrations" and thus concluded that "that p23 and FKBP51 are not acting independently but cooperatively in the modulation of tau aggregation". However, this is not clear from the data. When looking at figure 1C, the T_m of tau+p23 (1:0.1) is not significantly different from the one of tau+p23+FKBP51 (1:0.1,0.1). With higher dose, tau+p23 (1:0.2) and tau+p23+FKBP51 (1:0.2,0.2) might be different, but with a very small statistical margin. Similarly, when looking at the max ThT, the value of tau+ FKBP51 (1:0.1) is not significantly different from the one of tau+p23+FKBP51 (1:0.1,0.1). So, it rather indicates that the effect are merely additive, with FKBP51 increasing ThT intensity and p23 increasing T_m , and not cooperative.

Another concern is about the quantification of the aggregates. While ThT intensity increases by up to a factor of 3, the authors use a SDS-PAGE analysis to state that the amount of aggregate are equal or lower. The latter analysis has no replicate and shows very subtle differences between the conditions, which is not convincing for a semi-quantitative method such as SDS-PAGE analysis after fiber centrifugation. I would recommend to quantify the amount of aggregate in a more robust way. Furthermore, all aggregated samples show the appearance of 2 bands at lower MW (which then represent the major species in the wells) that were not present in the monomer wells. It might reveal proteolytic activity during the aggregation process. Could the author show a SDS-page gel of the pellet in addition to the supernatant to verify that the protein is intact?

- Section "FKBP51 binds to p23"

The authors emphasis a loss of intensity in the region 114-130. This is not clear at all from data presented in Fig 2b, where the intensity loss is throughout the sequence. Consequently, the inset highlight in red the region 114-130 as "strongly perturbed", which is not supported by the data. Especially, these residues show no CSP. In general, the inset is too small, and the other parts highlighted in red are not labelled with residue numbers.

In figure 3C, a significant number of new peaks appear in the spectrum. Could the author comment on their origin? The CSP seem to be small despite a 10-fold excess of the binding partner. Could the authors comment on it? Also, it would be nice to be more quantitative, representing for instance the proportion of TPR peaks that indeed shift and by how much in general.

- Section on the MEEVD domain.

The authors used a labelled p23 in complex with unlabeled FKBP51, onto which they added up 30 fold excess MEEVD, in order to derive that "p23 and MEEVD compete to bind with FKBP51 and demonstrates that their binding sites on the TPR

domain of FKBP51 are overlapping". Since the authors already mapped the binding site of p23 on FKBP51 using labelled FKBP51 (part just before), it is surprising that they do not simply mixed MEEVD with labelled FKBP51 to assess a potential competing binding site.

Then, a potential pitfall is that adding 30-fold excess of MEEVD, reaching an addition of almost 1mM, could have nonspecific effects leading to a recovery of the intensities from the p23 protein.

- Discussion

I find the Fig 7, lower pathway, misleading. The sketch essentially states that it is the complex FKBP51-P23 that binds to tau and has protective effect against aggregation. However, it is shown throughout the manuscript that both co-chaperones can bind tau independently. More importantly, Fig 1 show that the aggregation-preventing effecting mostly, if not solely, originates from p23. Thus, I recommend revising the sketch or providing more data supporting the need for a tertiary complex to interact with tau.

Minor points :

SAXS scattering data should be shown in figure 5 (not only in SI), overlapped with the fit used to extract the 3D model currently presented in panel A.

Reviewer #2

(Remarks to the Author)

The manuscript reports on structural interactions between tau and a chaperone complex comprised of p23 and FKBP51, combining a series of NMR, SAXS and mutagenesis experimental results with computational analysis including docking.

The major result is that the C terminal tail p23 recognizes the TPR (...) domain of FKBP51 and tau, and that the transient trimeric complex of the three proteins (p23, FKBP51 and tau) together inhibits tau aggregation and acts against the toxicity of Hsp90-FKBP51.

Since Hsp90 has highly promiscuous interactions with various receptors and kinases, the authors propose that co-chaperones of Hsp90 such as p23 and FKBP51 may be more feasible drug targets, justifying the significance of the study. One of the challenges in this area of study is that there are many cryo-EM structures of tau fibrils but the relationship between the intermediate complexes and the pathogenic tau structures is poorly understood.

p23 is 160 amino acids or 19 kDa and has a ~50 residue C terminal tail that is highly flexible disordered, which is essential for chaperone function and thereby putatively involved in interactions with tau but this has not been previously demonstrated.

FKBP51 is a 51 kDa neuronal protein and the tetratricopeptide repeat (TPR) domain binds to various heat shock proteins. (The authors could please define FK1, FK2 and TPR on page 2 where these terms are introduced.)

Fig 1 shows aggregation assays of tau with and without p23 and FKBP51. Under the conditions examined here, nucleation of tau alone initiates in ~14 hrs and fluorescence intensity indicative of fibril formation reaches a maximum at about 30 hrs. (Fig 1b could be improved by plotting the averages and some representative error bars perhaps, with the full set of data in supporting information; this would improve the readability.) p23 added to tau shifts the nucleation to longer time. FKBP51 shifts the data to a longer "span" as indicated on the y axis of Fig 1c. This axis relies upon interpretation of the overall fluorescence intensity, which is interpreted (speculatively) as arising due to a "different structure of the tau fibril or due to the presence of FKBP51 on the fibril surface". Based on SDS PAGE gels, the authors note that "a comparable amount of tau aggregated both in the presence and absence of FKBP51" (Supp Fig 1), so these are reasonable hypotheses to pursue. The analysis in Fig 1 would benefit from comparison with other types of dyes to understand if there is a structural interaction with thioflavin T. The conclusion here that p23 and FKBP51 "are not acting independently but cooperatively" is not convincing; note for example that the cases where the overall ratios of tau to others are 1:0.2 are not significantly different, such as tau:p23 1:0.2 (green) and tau:p23:FKBP51 1:0.1:0.1 (black). The one outlier is where equimolar amounts of p23 and FKBP51 were added, with a total of 0.4 fraction relative to tau; i.e., 1:0.2:0.2 ratios. The proper controls for comparison would be 1:0.4 ratios of tau:p23 and tau:FKBP51 as separate measurements. It is also necessary to use another dye or orthogonal means to quantify the rates since the authors recognize that there may be a systematic shift in the ThT fluorescence due to interactions with the chaperone proteins.

Fig 2 pursues the hypothesis that p23 and FKBP51 are interacting cooperatively. Solution NMR data here show shifts in the p23 spectra upon addition of FKBP51 at ratios of 1:1, 2:1, 3:1 FKBP51:p23. The p23 concentration is 40 micromolar (or 25 micromolar?) so the apparent interaction Kd in the high micromolar to low millimolar range, quite weak in context of the physiological concentrations of these proteins. Could the authors please comment on the significance of this Kd (and/or did they compute the Kd more precisely)? Also please clarify specifically for the data in Fig. 2, what are the concentrations, since on page 19 there are two separate conditions described, which are for the wild type and truncated construct. It would help to state this in the legend for Fig. 2. Overall the conclusion from Fig 2 is that there is a weak interaction arising from FKBP51 primarily impacting residues 80 to 110 of p23.

Fig 3 shows HMQC spectra of ¹³C labeled methyl groups for Val and Leu residues in FKBP51 upon addition of p23. The TPR domain signals get stronger upon addition of p23. What is the physical basis of this observation? Is the TPR domain in

intermediate exchange without p23 but then moved to slow exchange upon addition?

Fig 4 is a competition between p23 and a peptide (MEEVD) from Hsp90. Starting with the NMR signals of p23 in complex with FKBP51, where p23 signals are lost, very large excesses (10 to 30 fold) of MEEVD were added to recover up to about half of the signal intensity from p23. It is hard to be convinced of the specificity of this interaction from the relatively small effects observed even at these large ratios. The authors refer to data from another publication (ref 39) but it is not clear to this reviewer why they did not do the experiment with MEEVD added to labeled FKBP51 to observe the interactions with K272, E273, K352, N322, and K329? Also here some further quantification of the K_d values for the interactions would be helpful to understand if the in vitro conditions have relevance to physiological concentrations.

Fig 5 presents the model of the p23-FKBP51 complex based on HADDOCK calculations, starting from the crystal structures of the two proteins. Here again it is a little uncertain why the authors are using a combination of chemical shift perturbations but data from different conditions for the FKBP51 to MEEVD peptide interactions. Also were no intermolecular NOEs observable? Overall the descriptions in this section are not as clear as would be desirable, and the validation of the structural model with R88A and R93A mutations is quite a blunt tool. Another concern here is that the SAXS data do not appear to fit well from the presentation of Fig 5b and 5c. Some further discussion of the statistics of the modeling would be appropriate.

Fig 6 illustrates the data supporting formation of a trimeric complex. Again here the effects in both chemical shift perturbations and intensity changes are quite modest, despite very large excesses of p23 added. The authors note that the changes are most significant in the P2 region "that is the most positively charged region of tau", which raises the concern that the interaction is a non-specific electrostatic interaction arising from the large excess of p23 added to the buffer.

The discussion seems to overstate some of the results. e.g., page 14 last paragraph states that "p23 strongly delays tau's aggregation while FKBP51... did not slow tau's amyloid formation kinetics". In light of the concerns raised about the significance of Fig 1, this conclusion is not warranted. Again a few lines from the bottom of page 15, "p23 slows fibrillation kinetics of tau even at substoichiometric concentrations", and "our results further revealed that the presence of both p23 and FKBP51 attenuated tau's aggregation to a greater extent than p23 alone at comparable concentrations". But the concentrations differ by a factor of 2, as noted in the comments above.

In the end, then the modeled structure is based on relatively weak chemical shift perturbations, a competition measurement with a very large excess concentration of MEEVD, modeling from another structure under different conditions, and a lack of distance information in the form of NOEs or PREs.

In conclusion, then in this reviewer's assessment, there are some areas that could require further improvement and validation in order to make a more convincing case, and this study is not ready for publication yet.

Reviewer #3

(Remarks to the Author)

In the current study, authors investigated the impact and molecular links between Tau and Hsp90 and its co-chaperons p23 and FKBP-51 in Tau aggregation in Alzheimer's disease and other tauopathies. Authors used molecular docking and bioinformatics tools and discovered how p23 and FKBP-51 chaperons regulates tau aggregation in disease. If conformed by other independent groups. the current study results are truly groundbreaking.

Minor changes - please change the first sentence of abstract - 'the pathological deposition of amyloid beta and tau-phosphorylation are a major pathological hallmarks of Alzheimer's disease and other tauopathies'.

Any cell biology and/or protein-protein interaction data could strengthen the findings.

Reviewer #4

(Remarks to the Author)

Amyloid deposits of the microtubule-binding protein tau in patient's brains are associated with Alzheimer's disease and other tauopathies. In the deposits, tau is hyperphosphorylated and soluble tau oligomers are believed to be toxic to the cells.

Molecular chaperones including Hsp90 are believed to be modulators of tau toxic function, either by interfering with aggregation or regulating pathways leading to abnormal phosphorylation of tau.

Hsp90 with the assistance of a plethora of cochaperones enables the folding and maturation of important regulatory proteins such as nuclear receptors and protein kinases. It is believed to be one of the most abundant chaperones in the eukaryotic cell. Among the cochaperones are p23, which binds to N-terminal ATPase domain of Hsp90 and stabilizes it in the closed, ATP-bound state, and FKBP51, a peptidyl prolyl isomerase, which recognizes the C-terminal MEEVD motif of Hsp90 and assists the folding of client proteins. The reported dissociation constant for the latter complex is 174 nM (Pirkl & Buchner, 2001).

In the present manuscript, a complex between p23 and FKBP51 is proposed to delay tau amyloid formation.

In a tau amyloid formation assay, the mixture of p23 and FKBP51 seems to delay the half point marginally more than p23 alone, while FKBP51 alone has apparently no effect.

Using NMR titrations, the authors show that there is a binding interaction between p23 and FKBP51, but fail to determine a dissociation constant. To do this is essential for estimating whether meaningful quantities of such a complex would form in presence of high concentrations of Hsp90 in the cell. One could simply use ITC like in the paper above.

FKBP51 appears to interact with the folded domain of p23, but the flexible C-terminal tail of p23 is required for the interaction as well.

p23 interacts with the TPR domain in FKBP51, which harbors the binding site for the MEEVD peptide of Hsp90. The peptide competes with p23 binding.

Using protein docking, the authors create a model for the complex. The model is consistent with the p23-R88A/R93A mutant having decreased affinity to FKBP51.

Moreover, the author use SEC-SAXS to estimate the shape of the complex. Here it is unclear whether a stoichiometric complex was analyzed. The chromatogram suggests a considerable amount of free p23, despite substantial FKBP51 (both concentrations 175 μ M?). If p23 was strongly substoichiometric, it is questionable whether the differences in the shape models are meaningful. An analogous SEC experiment should be analyzed by an orthogonal method, e.g. SDS-PAGE, to determine the stoichiometry of the first band.

Finally, the author report NMR titrations to probe the binding interface between p23 and tau. In tau, the P2 region and the microtubule binding repeats seem to interact with p23, when the disordered C-terminal tail is present in p23. Tau seems to interact mostly with the C-terminal tail of p23. Based on this, a model for a ternary complex with FKBP51 is constructed.

I am very skeptical whether meaningful quantities of the p23:FKBP51 complex exist in neurons. The cellular concentrations of p23 and FKBP51 must be far below the concentrations in the NMR and SEC-MALS experiments. On top of that the abundant Hsp90 likely competes effectively with complex formation. This competition should be investigated experimentally to warrant the claims by the authors.

Moreover, the formation of a ternary p23-FKBP51-Tau complex that regulates tau aggregation, is actually not directly shown in the manuscript.

Detailed points:

What are the physiological concentrations of p23, FKBP51 and Hsp90 in human cells, ideally in neurons?

How does the tau aggregation delay by p23 and p23/FKBP51 compare to other molecular chaperones (J-domain proteins, Hsp70, Hsp90, small Hsps)?

In Fig. 1d the authors plot the T_m of Tau amyloid formation alone or in combination with the co-chaperones at different concentrations. They use a one-way ANOVA to compare the T_m of Tau aggregation alone or in combination of the co-chaperones. However, when comparing Tau:p23 1:0.2 and Tau:p23:FKBP51 1:0.2:0.2, they use a one-tailed unpaired t-test, when they must use the same one-way ANOVA as before, since ANOVA is a t-test with the necessary correction when doing multiple comparison. This should be corrected.

The difference in T_m between Tau:p23 1:0.2 and Tau:p23:FKBP51 1:0.2:0.2 is clearly quite small. Maybe if the authors show ThT curves normalized to the respective plateau fluorescence and quantifying the T_m in hours instead in days, these may help to show the difference between Tau:p23 and Tau:p23:FKBP51 more clearly.

Supplementary Fig. 1 How were the bands in this SDS-PAGE gel detected and quantified? Why are p23 and FKBP51 not visible where added? Is this a representative result? What is the error margin? Are the results significant? Does p23 also interact with fibrils?

It would be interesting to create a p23:FKBP51 complex model with AlphaFold2-multi. This would also take sequence conservation into account. Are the residues in the globular interface of p23 conserved?

Experimental proof for a ternary tau:p23:FKBP51 complex is missing and should be provided. This could be done by immune precipitation, native PAGE or size exclusion chromatography. If the complex is too transient, chemical crosslinking could be used to stabilize the complex.

Version 1:

Reviewer comments:

Reviewer #1

(Remarks to the Author)

The authors have addressed most of the technical concerns. The SAXS scattering data overlaid with the fit used to generate the 3D model are still missing.

However, the concerns raised by the reviewers about cooperativity is not seriously addressed. Despite a few wording changes, the manuscript, in its title, abstract and discussion, still largely conveys the message that the regulation of tau aggregation originates from the complex. In other words, from a cooperative action of the two factors. This is not supported by the data. The T_m from ThT do not show cooperativity. P23 alone delays aggregation. Adding the delay from FKBP51 alone and the delay from P23 alone is not significantly different from the delay triggered by both factors together. Thus, while the structural work is remarkable, the mechanisms of action in the context of tau aggregation and the associated pathologies is purely speculative at this point, if not in disagreement with the data.

Reviewer #2

(Remarks to the Author)

The authors have done a noteworthy job of revising the manuscript and responding in detail to the comments from my previous review, as well as those of the other reviewers. I find the responses overall to be quite convincing and consider the changes to the manuscript to be helpful in improving the clarity, completeness and scientific impact of the study. At this point I recommend publication.

Reviewer #4

(Remarks to the Author)

In their revision, the authors modified the text and added additional experiments. Overall, the revised manuscript is clearly improved. Most of my points of criticism have been adequately addressed in the revised manuscript. I am however still doubtful whether sufficient quantities of the p23-FKBP51 complex would form in cells to modify aggregation of tau and disease progression in tauopathies. The now determined dissociation constant of 67 μM for the p23-FKBP51 complex is pretty high and formation of complexes with Hsp90 dimers such as p23:(Hsp90)₂, FKBP51:(Hsp90)₂ and p23:FKBP51:(Hsp90)₂ should be preferred, judging from prior data. The presentation of the SAXS data is misleading. According to the response by the authors the FKBP51+p23 peak in the SEC chromatogram is mostly FKBP51 alone – most of the initially equimolar p23 has dissociated and elutes later (Fig. 5a). So, writing “The SAXS data of the p23-FKBP51 complex at an equimolar ratio generated a density map with R_g 40.8 Å and D_{max} 155.6 Å.” is not correct. The underlying scattering curve is from a mixture of species with a minor contribution of FKBP51:p23 to FKBP51 scattering. The scheme in Fig. 7 disregards the existence of (p23)₂:(Hsp90:ATP)₂ and p23:FKBP51:(Hsp90)₂ complexes, for both of which high-resolution structures are available. The new crosslinking data (Suppl. Fig. 8) are interesting, presenting evidence for a species containing FKBP51, p23 and tau. However, the analysis seems to be incomplete. There are many other bands representing crosslinking products in Suppl. Fig. 8a as well, while the most of p23 and FKBP51 do not seem to engage in intermolecular crosslinking. Control experiments exploring by SDS-PAGE the crosslink products of p23 alone, FKBP51 alone, p23/tau, FKBP51/tau and FKBP51/p23 would be insightful for the interpretation of the crosslink pattern from the ternary mixture. Western blot probing against the components would allow assigning the identity of the crosslinking products. Were there tau-tau crosslinks also observed, i. e. is the analyzed SDS-PAGE band a complex with monomeric or oligomeric tau? Are the observed FKBP51-p23 crosslinks compatible with the model from molecular docking?

Version 2:

Reviewer comments:

Reviewer #1

(Remarks to the Author)

The main comment in the last review round was on the lack of evidence for cooperativity between P23 and FKBP51 in regulating aggregation. Despite apparent acknowledgement of the comment in the response submitted, the first result is entitled “p23-FKBP51 cooperatively modulate tau aggregation”.

Figure 1 evaluate aggregation inhibition through ThT assays. It shows no effect of FKBP51 in both ratios of tau: FKBP51. Thus no claims should be made that FKBP51 inhibit aggregation. Such claim is present for example in the abstract “Here we reveal an Hsp90-independent modulation of tau aggregation by the two Hsp90 co-chaperones p23 and FKBP51 and their molecular complex”. I also don't understand the basis for the freshly added statement in the discussion “The PPlase activity of FKBP51 may enhance the aggregation of tau whereas the chaperone activity may delay tau's aggregation.”

In contrast P23 has a very significant effect on aggregation kinetics. The addition of FKBP51 at lower ratio (1:0.1:0.1) has no influence. The addition of FKBP51 to the higher ratio (1:0.2:0.2) has a small effect compare to P23. This is the only data that might point to cooperativity, but given the short statistical margin ($n=3$, $p=0.0366$) and the very small delaying effect compare to a very significant effect of P23 alone, it seems too weak to really claim a cooperativity effect. It rather looks like P23 alone is governing the aggregation delaying effect.

Reviewer #4

(Remarks to the Author)

The revised manuscript addresses most of my concerns.

The title of Fig. 7, "p23 and Hsp90 compete to form a complex with FKBP51." does not suggest that "a core element of the scheme is the protein tau", as stated by the authors in their response. Better revise to: "Competition of p23 and Hsp90 for FKBP51 modulates interactions with tau." or similar.

To refer the reader to the selective view of the FKBP51/p23/Hsp90/tau interactions in Fig. 7, the legend should also state: "Of note, binary p23-Hsp90 and ternary p23-FKBP51-Hsp90 complexes do also exist (not shown)."

Version 3:

Reviewer comments:

Reviewer #2

(Remarks to the Author)

The authors have effectively responded to the prior reviews and the manuscript is now ready for publication.

Reviewer #4

(Remarks to the Author)

I am happy with the present version of the manuscript.

Reviewer #1:

This article describes the interaction between the protein tau and two co-chaperone p23 and FKBP51. It mostly focuses on structural aspect of this interaction by identifying interaction sites. The report brings new data confirming direct interactions but does not provide breakthrough concepts or approaches to the field. It rather provides finer details on aggregation-modulating interactions with HSP90/FKBP51 that were already revealed, including by the same group a few years before (e.g.. Oroz et al. Nat Comm 2018).

Reply: We thank the reviewer for the careful evaluation of the manuscript. Please note that the current study provides a breakthrough concept as it (i) reveals a previously unknown direct, HSP90-independent interaction between FKBP51 and p23, and (ii) structurally characterizes this novel interaction. The breakthrough concept is that our current study indicates that the formation of a protective p23-FKBP51-tau trimeric complex (separate from Hsp90) may be a regulation mechanism to counteract toxic Hsp90-FKBP51 mediated toxicity (studied in Oroz et al. Nat Comm 2018).

The report is overall of good quality with some revision required as detailed below. One of the main conceptual concern is about the support for the notion of cooperativity as both binding and aggregation modulation occur without the need for the three partners. More details are provided below. - About the effect of chaperones on aggregation: The authors stated that “The presence of both co-chaperones attenuated the aggregation of tau to a greater extent than the individual co-chaperones at the comparable concentrations” and thus concluded that “that p23 and FKBP51 are not acting independently but cooperatively in the modulation of tau aggregation”. However, this is not clear from the data. When looking at figure 1C, the T_m of tau+p23 (1:0.1) is not significantly different from the one of tau+p23+FKBP51 (1:0.1,0.1). With higher dose, tau+p23 (1:0.2) and tau+p23+FKBP51 (1:0.2,0.2) might be different, but with a very small statistical margin. Similarly, when looking at the max ThT, the value of tau+ FKBP51 (1:0.1) is not significantly different from the one of tau+p23+FKBP51 (1:0.1,0.1). So, it rather indicates that the effect are merely additive, with FKBP51 increasing ThT intensity and p23 increasing T_m , and not cooperative.

Reply: We thank the reviewer for raising this point. Please note that the presence of FKBP51 has no significant effect on the T_m of aggregation at the lower concentrations used in our study (1:0.1 & 1:0.2). This is most probably because of the tug-of-war between the two different activities of FKBP51, the PPIase activity, and the chaperone activity. The PPIase activity of FKBP51 may enhance the aggregation of tau whereas the chaperone activity is expected to delay tau’s aggregation. Such opposing roles on the aggregation propensity of tau and alpha-synuclein were previously reported for the two PPIases FKBP12 and PPIA/CypA, respectively (ref #50,51). The two activities depend on the concentration of the chaperone/co-chaperone, with a lower concentration favoring PPIase activity while at higher concentrations the chaperone activity may dominate. Thus, the slight but statistically significant increase in the T_m of aggregation for tau+p23+FKBP51 (1:0.2:0.2) in comparison to tau + p23 (1:0.2) suggests that the effect on T_m may not be additive but synergistic.

To address the higher ThT intensity in the presence of FKBP51, we added the amyloid-binding dye curcumin to preformed fibrils (Supplementary Fig. 2). Notably, tau fibrils formed in the presence of FKBP51, but not p23, exhibited increased fluorescence upon curcumin addition compared to unmodified tau fibrils. Furthermore, the fluorescence intensity of fibrils formed with tau + p23 + FKBP51 (1:0.2:0.2) was significantly higher than that of fibrils formed with tau + FKBP51 (1:0.2), suggesting a synergistic effect rather than a mere additive one on fluorescence intensity.

However, we agree with the reviewer that when both p23 and FKBP51 are present it is difficult to clearly distinguish between synergistic and independent behaviour. We therefore adjusted the statements on page no. 5:

“The presence of both co-chaperones attenuated the aggregation of tau to a slightly larger extent than the individual co-chaperones at the comparable concentrations (Fig. 1c,d).”

“The combined data suggests that p23 and FKBP51 may act both independently and synergistically in modulating tau aggregation.”

Another concern is about the quantification of the aggregates. While ThT intensity increases by up to a factor of 3, the authors use a SDS-PAGE analysis to state that the amount of aggregate are equal or lower. The latter analysis has no replicate and shows very subtle differences between the conditions, which is not convincing for a semi-quantitative method such as SDS-PAGE analysis after fiber centrifugation. I would recommend to quantify the amount of aggregate in a more robust way.

Reply: We thank the reviewer for the suggestion. We repeated the SDS-PAGE experiment three times and observed that there is no significant difference between the % of aggregated proteins. Accordingly, we have updated the Supplementary Fig. 3 (revised version) as well as the result section on page no. 5: *“Notably, while the final fluorescence intensity was high even at the 1:0.2:0.2 molar ratio (Supplementary Fig. 1,2), quantification of the supernatant suggested a comparable amount of aggregated tau (Supplementary Fig. 3).”*

Furthermore, all aggregated samples show the appearance of 2 bands at lower MW (which then represent the major species in the wells) that were not present in the monomer wells. It might reveal proteolytic activity during the aggregation process. Could the author show a SDS-page gel of the pellet in addition to the supernatant to verify that the protein is intact?

Reply: The SDS -PAGE gel (shown below) of the pellet of tau fibril suggests that the protein is still intact.

Section “FKBP51 binds to p23”: The authors emphasis a loss of intensity in the region 114-130. This is not clear at all from data presented in Fig 2b, where the intensity loss is throughout the sequence. Consequently, the inset highlight in red the region 114-130 as “strongly perturbed”, which is not supported by the data. Especially, these residues show no CSP. In general, the inset is too small, and the other parts highlighted in red are not labelled with residue numbers.

Reply: Please note that the global decrease in the signal intensity of residues present in the structured domain (residue 1 to 110) of p23 is because of the increase in molecular weight due to p23-FKBP51 complex formation that leads to enhanced transverse relaxation. After the structured domain, residues within the initial segment of the flexible tail (111-115) exhibit a regain in intensity, followed by a subsequent decrease until residue 127, before returning to approximately 1 (Fig. 2b, top panel). Consequently, residues predicted to form an alpha-helix by AlphaFold2 demonstrate a distinct decrease in intensity compared to other residues within the C-terminal tail of p23 (Fig. 2b, top panel).

We thank the reviewer for pointing out the mistake in using the word “strongly perturbed” for residues 114-130. In the revised manuscript we have removed the word from the figure legend.

We have also updated Fig. 2 so that the inset is clearly visible and labelled the residue numbers on the structure of p23.

In figure 3C, a significant number of new peaks appear in the spectrum. Could the author comment on their origin? The CSP seem to be small despite a 10-fold excess of the binding partner. Could the authors comment on it? Also, it would be nice to be more quantitative, representing for instance the proportion of TPR peaks that indeed shift and by how much in general.

Reply: Please note that the new peaks observed in Fig. 3c represent peaks of p23 appearing at high concentrations due to the natural abundance of ¹³C. In the revised manuscript, we clarify this in the figure legend of Fig. 3c.

We have now determined the binding affinity between p23 and FKBP51 (67 μ M) by ITC (Supplementary Fig. 4). Thus, the CSP observed in the NMR experiments is in agreement with the binding affinity between the two proteins.

We have updated Fig. 3 with a quantitative determination of the CSPs for all Leu and Val residues of FKBP51.

- Section on the MEEVD domain: The authors used a labelled p23 in complex with unlabeled FKBP51, onto which they added up 30 fold excess MEEVD, in order to derive that “p23 and MEEVD compete to bind with FKBP51 and demonstrates that their binding sites on the TPR domain of FKBP51 are overlapping”. Since the authors already mapped the binding site of p23 on FKBP51 using labelled FKBP51 (part just before), it is surprising that they do not simply mixed MEEVD with labelled FKBP51 to assess a potential competing binding site. Then, a potential pitfall is that adding 30-fold excess of MEEVD, reaching an addition of almost 1mM, could have nonspecific effects leading to a recovery of the intensities from the p23 protein.

Reply: We thank the reviewer for the suggestion. To further elucidate the comparable binding site of MEEVD and p23 on FKBP51, we conducted 2D ^1H - ^{13}C methyl-TROSY spectra of ^{13}C -methyl labeled FKBP51 in the absence or presence of five-fold and fifteen-fold molar excesses of MEEVD peptide (Supplementary Fig. 6a). The addition of MEEVD caused perturbations to the same methyl resonances from the TPR domain of FKBP51 as those induced by p23 (Supplementary Fig. 6). Furthermore, additional resonances from the TPR domain were perturbed upon p23 addition (Supplementary Fig. 6c,d). In the revised manuscript, we have added these data in Supplementary Fig. 6 and discuss the results on page no. 9.

Please note that the 2D ^1H - ^{13}C methyl-TROSY experiments of ^{13}C -methyl labeled FKBP51 in the absence and presence of MEEVD were previously recorded by us (Supplementary Fig. 6a). Subsequently, we enhanced the spectral resolution by conducting 2D ^1H - ^{13}C HMQC spectra of ^{13}C -methyl labeled FKBP51 (Supplementary Fig. 6b). However, despite the improved resolution, there was no significant difference in the information obtained from the two experiments. Due to cost constraints and the need to adhere to the manuscript's timeline, we were unable to conduct similar 2D ^1H - ^{13}C HMQC spectra experiments for ^{13}C -methyl labeled FKBP51 in the absence or presence of MEEVD.

- Discussion: I find the Fig 7, lower pathway, misleading. The sketch essentially states that it is the complex FKBP51-P23 that binds to tau and has protective effect against aggregation. However, it is shown throughout the manuscript that both co-chaperones can bind tau independently. More importantly, Fig 1 show that the aggregation-preventing effecting mostly, if not solely, originates from p23. Thus, I recommend revising the sketch or providing more data supporting the need for a tertiary complex to interact with tau.

Reply: Please note that FKBP51 is reported to stimulate tau aggregation in-vivo (ref #33) and also in our in-vitro aggregation assay we didn't observe any delaying effect of FKBP51 on tau's aggregation. However, the p23-FKBP51 complex delays tau aggregation. Thus, the hypothesis behind Fig. 7 is that the p23-FKBP51 complex can exert a protective role in the toxicity mediated by FKBP51 alone on tau's aggregation. We agree with the reviewer that p23 alone also has protective role in regulating tau's aggregation and thus in the revised manuscript we have updated the Fig. 7 showing the protective role of the p23-tau complex. We have also added the following sentences at the end of the discussion section on page no. 18: “While p23 alone can exert a protective role in regulating tau aggregation (Fig. 1,7), the p23-FKBP51 complex can potentially counteract the FKBP51-mediated tau toxicity.”

Also, FKBP51 is reported to synergize with Hsp90 to promote toxic tau oligomerization by forming an Hsp90-FKBP51-tau pro-toxic complex (ref # 32,33). Thus, the formation of a protective

p23-FKBP51-tau trimeric complex may thus be a regulation mechanism to counteract toxic Hsp90-FKBP51-mediated toxicity.

In the revised manuscript in Supplementary Fig. 8, we further show the experimental validation of the formation of a p23-FKBP51-tau trimeric complex. To provide experimental validation for the formation of the FKBP51-p23-tau complex, we combined the three proteins at an equimolar ratio and stabilized the resulting trimeric complex by introducing the chemical crosslinker disuccinimidyl suberate (DSS). Upon loading the crosslinked complex onto an SDS-PAGE gel, a distinct band above 220 kDa emerged (Supplementary Fig. 8a). Subsequently, this band was excised and subjected to Mass Spectrometry analysis. The analysis unveiled a network of intermolecular crosslinks among p23, FKBP51, and tau, conclusively confirming the formation of the trimeric complex (Supplementary Fig. 8, Supplementary Table 1).

Minor points : SAXS scattering data should be shown in figure 5 (not only in SI), overlapped with the fit used to extract the 3D model currently presented in panel A.

Reply: In the revised manuscript, we have updated Fig. 5 accordingly.

Reviewer #2:

The manuscript reports on structural interactions between tau and a chaperone complex comprised of p23 and FKBP51, combining a series of NMR, SAXS and mutagenesis experimental results with computational analysis including docking. The major result is that the C terminal tail p23 recognizes the TPR (...) domain of FKBP51 and tau, and that the transient trimeric complex of the three proteins (p23, FKBP51 and tau) together inhibits tau aggregation and acts against the toxicity of Hsp90-FKBP51. Since Hsp90 has highly promiscuous interactions with various receptors and kinases, the authors propose that co-chaperones of Hsp90 such as p23 and FKBP51 may be more feasible drug targets, justifying the significance of the study. One of the challenges in this area of study is that there are many cryo-EM structures of tau fibrils but the relationship between the intermediate complexes and the pathogenic tau structures is poorly understood. p23 is 160 amino acids or 19 kDa and has a ~50 residue C terminal tail that is highly flexible disordered, which is essential for chaperone function and thereby putatively involved in interactions with tau but this has not been previously demonstrated.

Reply: We thank the reviewer for the careful evaluation of the manuscript and highlighting the relevance of the study.

FKBP51 is a 51 kDa neuronal protein and the tetratricopeptide repeat (TPR) domain binds to various heat shock proteins. (The authors could please define FK1, FK2 and TPR on page 2 where these terms are introduced.)

Reply: We thank the reviewer for the suggestion. In the revised manuscript, we defined the FK1, FK2, and TPR domains on page no. 2.

Fig 1 shows aggregation assays of tau with and without p23 and FKBP51. Under the conditions examined here, nucleation of tau alone initiates in ~14 hrs and fluorescence intensity indicative of fibril formation reaches a maximum at about 30 hrs. (Fig 1b could be improved by plotting the averages and some representative error bars perhaps, with the full set of data in supporting information; this would improve the readability.)

Reply: We thank the reviewer for the suggestion. To improve the readability, in the revised manuscript we now show the normalized fluorescence intensity curves in Fig. 1b. The non-normalized curves are shown in Supplementary Fig. 1.

p23 added to tau shifts the nucleation to longer time. FKBP51 shifts the data to a longer "span" as indicated on the y axis of Fig 1c. This axis relies upon interpretation of the overall fluorescence intensity, which is interpreted (speculatively) as arising due to a "different structure of the tau fibril or due to the presence of FKBP51 on the fibril surface". Based on SDS PAGE gels, the authors note that "a comparable amount of tau aggregated both in the presence and absence of FKBP51" (Supp Fig 1), so these are reasonable hypotheses to pursue. The analysis in Fig 1 would benefit from comparison with other types of dyes to understand if there is a structural interaction with ThT.

Reply: To understand if there is any structural interaction with ThT, we added the amyloid-binding dye curcumin to preformed fibrils (Supplementary Fig. 2). Notably, tau fibrils formed in the presence of FKBP51, but not p23, exhibited increased fluorescence upon curcumin addition compared to unmodified tau fibrils. Furthermore, the fluorescence intensity of fibrils formed with tau + p23 + FKBP51 (1:0.2:0.2) was significantly higher than that of fibrils formed with tau + FKBP51 (1:0.2). Thus, we have observed a similar trend in the fluorescence intensity for both dyes - ThT and curcumin.

The conclusion here that p23 and FKBP51 "are not acting independently but cooperatively" is not convincing; note for example that the cases where the overall ratios of tau to others are 1:0.2 are not significantly different, such as tau:p23 1:0.2 (green) and tau:p23:FKBP51 1:0.1:0.1 (black). The one outlier is where equimolar amounts of p23 and FKBP51 were added, with a total of 0.4 fraction relative to tau; i.e., 1:0.2:0.2 ratios. The proper controls for comparison would be 1:0.4 ratios of tau:p23 and tau:FKBP51 as separate measurements. It is also necessary to use another dye or orthogonal means to quantify the rates since the authors recognize that there may be a systematic shift in the ThT fluorescence due to interactions with the chaperone proteins.

Reply: Please note that the presence of FKBP51 has no significant effect on the T_m of aggregation at the lower concentrations used in our study (1:0.1 & 1:0.2). This is most probably because of the tug-of-war between the two different activities of FKBP51, the PPIase activity, and the chaperone activity. The PPIase activity of FKBP51 may enhance the aggregation of tau whereas the chaperone activity is expected to delay tau's aggregation. Such opposing roles on the aggregation propensity of tau and alpha-synuclein were reported for the two PPIases FKBP12 and PPIA/CypA, respectively (ref #50,51). The two activities depend on the concentration of the chaperone/co-chaperone, with a lower concentration favoring PPIase activity while at higher concentrations the chaperone activity may dominate. Thus, the slight but statistically significant increase in the T_m of aggregation for tau+p23+FKBP51 (1:0.2:0.2) in comparison to tau + p23 (1:0.2) suggests that the effect on T_m may not be additive but synergistic.

To address the higher ThT intensity in the presence of FKBP51, we added amyloid-binding dye curcumin to preformed fibrils (Supplementary Fig. 2). Notably, tau fibrils formed in the presence of FKBP51, but not p23, exhibited increased fluorescence upon curcumin addition compared to unmodified tau fibrils. Furthermore, the fluorescence intensity of fibrils formed with tau + p23 + FKBP51 (1:0.2:0.2) was significantly higher than that of fibrils formed with tau + FKBP51 (1:0.2), suggesting a synergistic effect rather than a mere additive one on fluorescence intensity.

However, we agree with the reviewer that when both p23 and FKBP51 are present it is difficult to clearly distinguish between synergistic and independent behaviour. We therefore adjusted the statements on page no. 5: *"The combined data suggests that p23 and FKBP51 may act both independently and synergistically in modulating tau aggregation."*

Regarding the suggested control experiment of tau:FKBP51 (1:0.4): this unfortunately cannot be done reliably, because of the low stability of FKBP51 at higher concentrations in our aggregation assay condition. We make this limitation in the discussion section of our manuscript clear (page no. 16): *"Currently, we cannot further investigate this hypothesis because of the lower stability of FKBP51 at higher concentrations in our aggregation assay."*

Fig 2 pursues the hypothesis that p23 and FKBP51 are interacting cooperatively. Solution NMR data here show shifts in the p23 spectra upon addition of FKBP51 at ratios of 1:1, 2:1, 3:1 FKBP51:p23. The p23 concentration is 40 micromolar (or 25 micromolar?) so the apparent interaction Kd in the high micromolar to low millimolar range, quite weak in context of the physiological concentrations of these proteins. Could the authors please comment on the significance of this Kd (and/or did they compute the Kd more precisely)?

Reply: We thank the reviewer for this suggestion. We have now determined the binding affinity between p23 and FKBP51 by ITC. The binding affinity between the two proteins is ~67 μ M. The data has been added in Supplementary Fig. 4 and discussed in the result section on page no. 5.

Also please clarify specifically for the data in Fig. 2, what are the concentrations, since on page 19 there are two separate conditions described, which are for the wild type and truncated construct. It would help to state this in the legend for Fig. 2. Overall the conclusion from Fig 2 is that there is a weak interaction arising from FKBP51 primarily impacting residues 80 to 110 of p23.

Reply: We thank the reviewer for this comment. Please note that the concentration of 15 N-labeled p23 WT was 40 μ M and the concentration of C-terminal deletion mutant of p23, i.e., 15 N-labeled p23 (1-119) was 25 μ M. We mention these values on page no. 21 in the method section entitled 'NMR Spectroscopy'.

Fig 3 shows HMQC spectra of 13 C labeled methyl groups for Val and Leu residues in FKBP51 upon addition of p23. The TPR domain signals get stronger upon addition of p23. What is the physical basis of this observation? Is the TPR domain in intermediate exchange without p23 but then moved to slow exchange upon addition?

Reply: We did not find any evidence that the signals of the resonances from the TPR domain get stronger upon addition of p23. Instead, we observed some additional peaks in the spectra. The new peaks observed in Fig. 3c represent peaks of p23 appearing at high concentrations due to the natural abundance of 13 C. In the revised manuscript, we clarify this in the legend to Fig. 3c.

Fig 4 is a competition between p23 and a peptide (MEEVD) from Hsp90. Starting with the NMR signals of p23 in complex with FKBP51, where p23 signals are lost, very large excesses (10 to 30 fold) of MEEVD were added to recover up to about half of the signal intensity from p23. It is hard to be convinced of the specificity of this interaction from the relatively small effects observed even at these large ratios. The authors refer to data from another publication (ref 39) but it is not clear to this reviewer why they did not do the experiment with MEEVD added to labeled FKBP51 to observe the interactions with K272, E273, K352, N322, and K329? Also here some further quantification of the Kd values for the interactions would be helpful to understand if the in vitro conditions have relevance to physiological concentrations.

Reply: We thank the reviewer for the suggestion. To further elucidate the comparable binding site of MEEVD and p23 on FKBP51, we conducted 2D 1 H- 13 C methyl-TROSY spectra of 13 C-methyl labeled FKBP51 in the absence or presence of five-fold and fifteen-fold molar excesses of MEEVD peptide (Supplementary Fig. 6a). The addition of MEEVD caused perturbations to the same methyl resonances from the TPR domain of FKBP51 as those induced by p23 (Supplementary Fig. 6). Furthermore, additional resonances from the TPR domain were perturbed upon p23 addition (Supplementary Fig. 6c,d). In the revised manuscript, we have added this data in Supplementary Fig. 6 and discuss the results on page no. 9.

Please note that the 2D 1 H- 13 C methyl-TROSY experiments of 13 C-methyl labeled FKBP51 in the absence and presence of MEEVD were previously recorded by us (Supplementary Fig. 6a). Subsequently, we enhanced the spectral resolution by conducting 2D 1 H- 13 C HMQC spectra of 13 C-methyl labeled FKBP51 (Supplementary Fig. 6b). However, despite the improved resolution, there was

no significant difference in the information obtained from the two experiments. Due to cost constraints and the need to adhere to the manuscript's timeline, we were unable to conduct similar 2D ^1H - ^{13}C HMQC spectra experiments for ^{13}C -methyl labeled FKBP51 in the absence or presence of MEEVD.

Fig 5 presents the model of the p23-FKBP51 complex based on HADDOCK calculations, starting from the crystal structures of the two proteins. Here again it is a little uncertain why the authors are using a combination of chemical shift perturbations but data from different conditions for the FKBP51 to MEEVD peptide interactions. Also were no intermolecular NOEs observable? Overall the descriptions in this section are not as clear as would be desirable, and the validation of the structural model with R88A and R93A mutations is quite a blunt tool. Another concern here is that the SAXS data do not appear to fit well from the presentation of Fig 5b and 5c. Some further discussion of the statistics of the modeling would be appropriate.

Reply: Please note that we couldn't assign the TPR domain of FKBP51 due to the limited solubility of the domain when expressed alone. Thus, we couldn't perform NOE or other experiments to support the structure determination of the p23-FKBP51 complex.

To model the structure of the p23-FKBP51 complex, identification of the active residues within both proteins is crucial. In p23, residues exhibiting the highest CSPs (11, 56, 88, 94, 103, and 107) were designated as the active site. However, direct information regarding the specific residues within the TPR domain interacting with p23 was unavailable due to incomplete assignment. Nevertheless, insights gained from competitive binding experiments suggested a shared binding site for both p23 and the MEEVD peptide on FKBP51. Thus, the residues implicated in MEEVD interaction were assigned as active residues on FKBP51, guiding the structural modeling of the FKBP51-p23 complex.

Regarding the SAXS data, please refer to page no. 13 of the manuscript-
"The SAXS data of the p23-FKBP51 complex at an equimolar ratio generated a density map with R_g 40.8 Å and D_{max} 155.6 Å. The increase in R_g and D_{max} is in agreement with the complex formation between FKBP51 and p23 (Fig. 5b-e). Further comparison of the density map of the free FKBP51 and p23-FKBP51 complex revealed additional electron density near the TPR domain of FKBP51 in the complex state (Fig. 5c). The complex structure of p23-FKBP51 (Fig. 4d) also fits into the density map calculated from SAXS although the disordered tail of FKBP51, and potentially also the C-terminal helix of FKBP51, may undergo conformational changes (Fig. 5e).

Additionally, we observed density near the FK2 and TPR domain of FKBP51 for the p23-FKBP51 complex. This additional density may be attributed to the presence of the 50 residue-long C-terminal tail of p23 that is missing in the crystal structure of free p23 (Fig. 5e). Together with the contribution of the disordered C-terminal residues of p23 to p23-FKBP51 complex formation revealed by NMR spectroscopy (Fig. 2b,c), the data suggest binding of the C-terminal tail of p23 to the FK2/TPR interface region in the p23-FKBP51 complex."

Fig 6 illustrates the data supporting formation of a trimeric complex. Again here the effects in both chemical shift perturbations and intensity changes are quite modest, despite very large excesses of p23 added. The authors note that the changes are most significant in the P2 region "that is the most positively charged region of tau", which raises the concern that the interaction is a non-specific electrostatic interaction arising from the large excess of p23 added to the buffer.

Reply: To provide experimental validation for the formation of the FKBP51-p23-tau complex, we combined the three proteins at an equimolar ratio and stabilized the resulting trimeric complex by introducing the chemical crosslinker disuccinimidyl suberate (DSS). Upon loading the crosslinked complex onto an SDS-PAGE gel, a distinct band above 220 kDa emerged (Supplementary Fig. 8a). Subsequently, this band was excised and subjected to Mass Spectrometry analysis. The analysis unveiled a network of intermolecular crosslinks among p23, FKBP51, and tau, conclusively confirming the formation of the trimeric complex (Supplementary Fig. 8, Supplementary Table 1).

Please note that we also see a clear effect on the aggregation kinetics of tau upon the addition of a sub-stoichiometric amount of p23 supporting the relevance of the p23-tau interaction.

The discussion seems to overstate some of the results. e.g., page 14 last paragraph states that "p23 strongly delays tau's aggregation while FKBP51... did not slow tau's amyloid formation kinetics". In light of the concerns raised about the significance of Fig 1, this conclusion is not warranted. Again a few lines from the bottom of page 15, "p23 slows fibrillization kinetics of tau even at sub-stoichiometric concentrations", and "our results further revealed that the presence of both p23 and FKBP51 attenuated tau's aggregation to a greater extent than p23 alone at comparable concentrations". But the concentrations differ by a factor of 2, as noted in the comments above. In the end, then the modeled structure is based on relatively weak chemical shift perturbations, a competition measurement with a very large excess concentration of MEEVD, modeling from another structure under different conditions, and a lack of distance information in the form of NOEs or PREs. In conclusion, then in this reviewer's assessment, there are some areas that could require further improvement and validation in order to make a more convincing case, and this study is not ready for publication yet.

Reply: We thank the reviewer for the useful suggestions and believe to have addressed them in the revised manuscript (please see our replies above).

Reviewer #3 :

In the current study, authors investigated the impact and molecular links between Tau and Hsp90 and its co-chaperons p23 and FKBP-51 in Tau aggregation in Alzheimer's disease and other tauopathies. Authors used molecular docking and bioinformatics tools and discovered how p23 and FKBP-51 chaperons regulates tau aggregation in disease. If conformed by other independent groups. the current study results are truly groundbreaking.

Reply: We thank the reviewer for the encouraging comments and highlighting the relevance of the study.

Minor changes - please change the first sentence of abstract - 'the pathological deposition of amyloid beta and tau-phosphorylation are a major pathological hallmarks of Alzheimer's disease and other tauopathies'.

Reply: We have made the following changes to the abstract: *"The pathological deposition of tau and amyloid-beta into insoluble amyloid fibrils are pathological hallmarks of Alzheimer's disease."*

Any cell biology and/or protein-protein interaction data could strengthen the findings.

Reply: We agree with the reviewer that cell biology experiments would strengthen the findings. We thus hope that our study will motivate other research groups to study the impact of the p23-FKBP51-tau interplay in cells.

Reviewer #4:

Amyloid deposits of the microtubule-binding protein tau in patient's brains are associated with Alzheimer's disease and other tauopathies. In the deposits, tau is hyperphosphorylated and soluble tau oligomers are believed to be toxic to the cells. Molecular chaperones including Hsp90 are believed to be modulators of tau toxic function, either by interfering with aggregation or regulating pathways leading to abnormal phosphorylation of tau. Hsp90 with the assistance of a plethora of co-chaperones enables the folding and maturation of important regulatory proteins such as nuclear

receptors and protein kinases. It is believed to be one of the most abundant chaperones in the eukaryotic cell. Among the cochaperones are p23, which binds to N-terminal ATPase domain of Hsp90 and stabilizes it in the closed, ATP-bound state, and FKBP51, a peptidyl prolyl isomerase, which recognizes the C-terminal MEEVD motif of Hsp90 and assists the folding of client proteins. The reported dissociation constant for the latter complex is 174 nM (Pirkl & Buchner, 2001). In the present manuscript, a complex between p23 and FKBP51 is proposed to delay tau amyloid formation. In a tau amyloid formation assay, the mixture of p23 and FKBP51 seems to delay the half point marginally more than p23 alone, while FKBP51 alone has apparently no effect.

Reply: We thank the reviewer for the careful evaluation of the manuscript.

Using NMR titrations, the authors show that there is a binding interaction between p23 and FKBP51, but fail to determine a dissociation constant. To do this is essential for estimating whether meaningful quantities of such a complex would form in presence of high concentrations of Hsp90 in the cell. One could simply use ITC like in the paper above.

Reply: We thank the reviewer for the suggestion. For the revised manuscript, we determined the binding affinity between p23 and FKBP51 by ITC. The binding affinity between the two proteins is around 67 μ M (Supplementary Fig. 4), as discussed in the result section on page no. 5.

FKBP51 appears to interact with the folded domain of p23, but the flexible C-terminal tail of p23 is required for the interaction as well. p23 interacts with the TPR domain in FKBP51, which harbors the binding site for the MEEVD peptide of Hsp90. The peptide competes with p23 binding. Using protein docking, the authors create a model for the complex. The model is consistent with the p23-R88A/R93A mutant having decreased affinity to FKBP51. Moreover, the author use SEC-SAXS to estimate the shape of the complex. Here it is unclear whether a stoichiometric complex was analyzed. The chromatogram suggests a considerable amount of free p23, despite substantial FKBP51 (both concentrations 175 μ M?). If p23 was strongly substoichiometric, it is questionable whether the differences in the shape models are meaningful. An analogous SEC experiment should be analyzed by an orthogonal method, e.g. SDS-PAGE, to determine the stoichiometry of the first band.

Reply: We thank the reviewer for raising the point. To perform the SEC-SAXS experiment, 150 μ M of FKBP51 was mixed with 150 μ M of p23. We agree with the reviewer that there is a substantial amount of unbound p23 present. However, the additional density in the FKBP51-p23 sample is not because of the presence of unbound p23 in the sample. This is because, during the analysis of the SEC-SAXS experiment, we can select the region (by selecting the frame numbers) for which the shape needs to be determined. We have selected the region of the complex peak and then processed the data to determine the shape. Thus, the additional density should arise because of the presence of p23 on the surface of FKBP51.

Finally, the author report NMR titrations to probe the binding interface between p23 and tau. In tau, the P2 region and the microtubule binding repeats seem to interact with p23, when the disordered C-terminal tail is present in p23. Tau seems to interact mostly with the C-terminal tail of p23. Based on this, a model for a ternary complex with FKBP51 is constructed. I am very skeptical whether meaningful quantities of the p23:FKBP51 complex exist in neurons. The cellular concentrations of p23 and FKBP51 must be far below the concentrations in the NMR and SEC-MALS experiments. On top of that the abundant Hsp90 likely competes effectively with complex formation. This competition should be investigated experimentally to warrant the claims by the authors. Moreover, the formation of a ternary p23-FKBP51-Tau complex that regulates tau aggregation, is actually not directly shown in the manuscript. Detailed points: What are the physiological concentrations of p23, FKBP51 and Hsp90 in human cells, ideally in neurons?

Reply: We thank the reviewer for these comments. To our knowledge, the concentration of p23 and FKBP51 in neurons is not accurately known. However, the knockdown of p23 and FKBP51 leads to a strong reduction of tau levels in cells suggesting that these co-chaperones can regulate tau biology in an Hsp90-independent manner (ref #1,7). We agree with the reviewer that it will be important to show whether FKBP51 and p23 can interact in cells in the presence of Hsp90 and we are convinced that the current in-vitro study (including the structure-based identification of mutants that block p23/FKBP51 complex formation) establishes an important basis to do so. Such studies are however beyond the scope of the current manuscript, in particular to investigate the p23/FKBP51 interaction independent of Hsp90.

To provide direct evidence for the formation of a ternary p23-FKBP51-Tau complex, we combined the three proteins at an equimolar ratio and stabilized the resulting trimeric complex by introducing the chemical crosslinker disuccinimidyl suberate (DSS). Upon loading the crosslinked complex onto an SDS-PAGE gel, a distinct band above 220 kDa emerged (Supplementary Fig. 8a). Subsequently, this band was excised and subjected to Mass Spectrometry analysis. The analysis unveiled a network of intermolecular crosslinks among p23, FKBP51, and tau, conclusively confirming the formation of the trimeric complex (Supplementary Fig. 8, Supplementary Table 1).

How does the tau aggregation delay by p23 and p23/FKBP51 compare to other molecular chaperones (J-domain proteins, Hsp70, Hsp90, small Hsps)?

Reply: We thank the reviewer for this suggestion. Because the current study focuses on p23/FKBP51 and we do not have all the other molecular chaperones available in the lab, we believe that addressing this question would be beyond the scope of the manuscript. Notably, a direct comparison of the different chaperones under identical assay conditions would be required to make reliable conclusions.

In Fig. 1d the authors plot the T_m of Tau amyloid formation alone or in combination with the co-chaperones at different concentrations. They use a one-way ANOVA to compare the T_m of Tau aggregation alone or in combination of the co-chaperones. However, when comparing Tau:p23 1:0.2 and Tau:p23:FKBP51 1:0.2:0.2, they use a one-tailed unpaired t-test, when they must use the same one-way ANOVA as before, since ANOVA is a t-test with the necessary correction when doing multiple comparison. This should be corrected.

Reply: We thank the reviewer for pointing this out. We have performed the statistical analysis between tau:p23 = 1:0.2 and tau:p23:FKBP51 = 1:0.2:0.2 using one-way ANOVA analysis (*p = 0.0366). In the revised manuscript, we have updated the figure legend of Fig. 1d accordingly.

The difference in T_m between Tau:p23 1:0.2 and Tau:p23:FKBP51 1:0.2:0.2 is clearly quite small. Maybe if the authors show ThT curves normalized to the respective plateau fluorescence and quantifying the T_m in hours instead in days, these may help to show the difference between Tau:p23 and Tau:p23:FKBP51 more clearly.

Reply: We thank the reviewer for the suggestion. To improve the readability, in the revised manuscript we now show the normalized fluorescence intensity curves in Fig. 1b. The non-normalized curves are shown in Supplementary Fig. 1. We didn't change the representation of T_m in hours from days as it did not make a strong difference.

Supplementary Fig. 1 How were the bands in this SDS-PAGE gel detected and quantified? Why are p23 and FKBP51 not visible where added? Is this a representative result? What is the error margin? Are the results significant? Does p23 also interact with fibrils?

Reply: Please note that the bands in the SDS-PAGE gels were quantified by ImageJ software. Briefly, the intensity of all the bands observed in one lane (for supernatants of different aggregated samples) was added and then the total intensity was divided by the intensity of monomeric tau.

We don't see the bands of p23 and FKBP51 in the supernatants because they were present in substoichiometric amounts and we loaded a small volume of supernatants on the SDS-PAGE gel to properly quantify the intensities.

We repeated the SDS-PAGE experiment three times and observed that there is no significant difference between the % of aggregated proteins. Accordingly, we have updated the Supplementary Fig. 3 (revised version) as well as the result section on page no. 5:

*“Notably, while the final fluorescence intensity was high even at the 1:0.2:0.2 molar ratio (Supplementary Fig. 1,2), quantification of the supernatant suggested a **comparable amount of aggregated tau** (Supplementary Fig. 3).”*

We loaded the pellets of tau aggregated in the absence or presence of both p23 & FKBP51 in the SDS-PAGE gel (shown below) and observed a band of p23 suggesting that it interacts with the fibrils. We also observed a band near 50 kDa, suggesting the potential interaction of FKBP51 with the tau fibrils. However, as the FKBP51 and tau run very close in the SDS-PAGE gel, we can't say surely if the observed band near 50 kDa is of FKBP51.

It would be interesting to create a p23:FKBP51 complex model with AlphaFold2-multi. This would also take sequence conservation into account. Are the residues in the globular interface of p23 conserved?

Reply: We thank the reviewer for the suggestion. Unfortunately, the p23:FKBP51 complex model predicted by AlphaFold2-multi doesn't match the experimental evidence. Yes, the residues in the globular interface of p23 are conserved.

Figure: Ensemble of complex structures predicted by AlphaFold2 for the interaction between FKBP51 & p23. In most of the structures, p23 is predicted to interact with the FK1 & FK2 domain of FKBP51.

Experimental proof for a ternary tau:p23:FKBP51 complex is missing and should be provided. This could be done by immune precipitation, native PAGE or size exclusion chromatography. If the complex is too transient, chemical crosslinking could be used to stabilize the complex.

Reply: We thank the reviewer for the suggestion. In the revised manuscript in Supplementary Fig. 8, we show the experimental validation of the formation of a p23-FKBP51-tau trimeric complex. To provide experimental validation for the formation of the FKBP51-p23-tau complex, we combined the three proteins at an equimolar ratio and stabilized the resulting trimeric complex by introducing the chemical crosslinker disuccinimidyl suberate (DSS). Upon loading the crosslinked complex onto an SDS-PAGE gel, a distinct band above 220 kDa emerged (Supplementary Fig. 8a). Subsequently, this band was excised and subjected to Mass Spectrometry analysis. The analysis unveiled a network of intermolecular crosslinks among p23, FKBP51, and tau, conclusively confirming the formation of the trimeric complex (Supplementary Fig. 8, Supplementary Table 1).

Reviewer #1:

The authors have addressed most of the technical concerns. The SAXS scattering data overlaid with the fit used to generate the 3D model are still missing.

Reply: We thank the reviewer for raising this point. We have included the SAXS scattering data and the fit used to generate the 3D model in Fig. 5c,d.

However, the concerns raised by the reviewers about cooperativity is not seriously addressed. Despite a few wording changes, the manuscript, in its title, abstract and discussion, still largely conveys the message that the regulation of tau aggregation originates from the complex. In other words, from a cooperative action of the two factors. This is not supported by the data. The T_m from ThT do not show cooperativity. P23 alone delays aggregation. Adding the delay from FKBP51 alone and the delay from P23 alone is not significantly different from the delay triggered by both factors together. Thus, while the structural work is remarkable, the mechanisms of action in the context of tau aggregation and the associated pathologies is purely speculative at this point, if not in disagreement with the data.

Reply: We thank the reviewer for allowing us to clarify the implications of our structural work. In the revised version of the manuscript, we have updated the title of the manuscript to: *“Interplay of p23, FKBP51, and their chaperone complex in regulating tau aggregation”*. Additionally, in the abstract we have updated the following sentence: *“Here we reveal an Hsp90-independent modulation of tau aggregation by the two Hsp90 co-chaperones p23 and FKBP51 and their molecular complex.”* In the discussion section, we now discuss the individual roles played by p23, FKBP51 in regulating tau aggregation:

“Using our recently developed co-factor-free aggregation assay of tau, we showed that the Hsp90 co-chaperone p23 strongly delays tau’s aggregation while FKBP51, another Hsp90 co-chaperone, did not slow tau’s amyloid formation kinetics (Fig. 1). While further studies are required, a tug-of-war between two different activities of FKBP51 may be present, the PPIase activity of the FK1 domain and the chaperone activity of FKBP51’s TPR domain. The PPIase activity of FKBP51 may enhance the aggregation of tau whereas the chaperone activity may delay tau’s aggregation. Such opposing roles on the aggregation propensity of tau and α -synuclein were reported for the two PPIases FKBP12 and PPIA/CypA, respectively^{50,51}. The two activities depend on the concentration of the chaperone/co-chaperone, with a lower concentration favoring PPIase activity while at higher concentrations the chaperone activity may dominate^{50,51}. Currently, we cannot further investigate this hypothesis because of the lower stability of FKBP51 at higher concentrations in our aggregation assay. Notably, FKBP51 stimulates tau aggregation in vivo³³, suggesting that its PPIase activity may play an important role in vivo.

In contrast to FKBP51, p23 slows the fibrillization kinetics of tau even at substoichiometric concentrations (Fig. 1). p23’s aggregation inhibition activity can be attributed to the interaction of the negatively charged C-terminal tail of p23 with tau’s positively charged aggregation-prone repeat domain (Fig. 6a-d). The C-terminal tail of p23 has previously been suggested to exert chaperone activity²⁶. Our results further revealed that the presence of both p23 and FKBP51 attenuated tau’s aggregation to a slightly larger extent than p23 alone at comparable concentrations (Fig. 1). This may arise from the formation of a trimeric complex between tau, p23, and FKBP51 in which tau’s repeat-domain interacts with the C-terminal tail of p23 and tau’s proline-rich domain clusters at the FK1 domain of FKBP51 (Fig. 6e, Supplementary Fig. 8). The trimeric p23-FKBP51-tau interaction is most likely not a single rigid complex, but rather a dynamic ensemble of trimeric complex structures in which different positively charged regions of Tau bind to the negatively charged C-terminal tail of p23.”

Reviewer #2:

The authors have done a noteworthy job of revising the manuscript and responding in detail to the comments from my previous review, as well as those of the other reviewers. I find the responses overall to be quite convincing and consider the changes to the manuscript to be helpful in improving the clarity, completeness and scientific impact of the study. At this point I recommend publication.

Reply: We thank the reviewer for the positive feedback.

Reviewer #4:

In their revision, the authors modified the text and added additional experiments. Overall, the revised manuscript is clearly improved. Most of my points of criticism have been adequately addressed in the revised manuscript.

Reply: We thank the reviewer for the positive feedback.

I am however still doubtful whether sufficient quantities of the p23-FKBP51 complex would form in cells to modify aggregation of tau and disease progression in tauopathies. The now determined dissociation constant of 67 μ M for the p23-FKBP51 complex is pretty high and formation of complexes with Hsp90 dimers such as p23:(Hsp90)₂, FKBP51:(Hsp90)₂ and p23:FKBP51:(Hsp90)₂ should be preferred, judging from prior data.

Reply: We agree with the reviewer that cell biology experiments would strengthen the findings. While we cannot make any statement regarding the abundance of the p23-FKBP51 complex in cells, our finding demonstrates that p23 can bind to FKBP51 providing a competitive interaction that can further regulate the p23/FKBP51/HSP90 interplay. We thus hope that our study will motivate other research groups to perform additional studies to further investigate this question in cells.

The presentation of the SAXS data is misleading. According to the response by the authors the FKBP51+p23 peak in the SEC chromatogram is mostly FKBP51 alone – most of the initially equimolar p23 has dissociated and elutes later (Fig. 5a). So, writing “The SAXS data of the p23-FKBP51 complex at an equimolar ratio generated a density map with R_g 40.8 Å and D_{max} 155.6 Å.” is not correct. The underlying scattering curve is from a mixture of species with a minor contribution of FKBP51:p23 to FKBP51 scattering.

Reply: We thank the reviewer for raising this point. Accordingly, we have updated the result section on page no. 13: *“Although an equimolar ratio of p23 and FKBP51 were injected into the SEC column, most of the p23 remained unbound (Fig. 5a). Thus, the scattering data for the FKBP51-p23 complex represents a mixture of species, with a minor contribution from the FKBP51-p23 complex to the overall FKBP51 scattering (Fig. 5g).”*

The scheme in Fig. 7 disregards the existence of (p23)₂:(Hsp90:ATP)₂ and p23:FKBP51:(Hsp90)₂ complexes, for both of which high-resolution structures are available.

Reply: We thank the reviewer for this suggestion. Please note that a core element of Fig. 7 is the protein tau. The role played by the (p23)₂:(Hsp90)₂ and p23:FKBP51:(Hsp90)₂ complexes in regulating tau aggregation remains however unknown, presenting a potential avenue for future study. Additionally, p23 promotes the closure of the Hsp90 dimer, resulting in a closed conformation for both the (p23)₂:(Hsp90)₂ and p23:FKBP51:(Hsp90)₂ complexes. In this closed conformation of Hsp90, the interaction with the client tau may change and thus we believe discussing this falls beyond the scope of the current manuscript.

The new crosslinking data (Suppl. Fig. 8) are interesting, presenting evidence for a species containing FKBP51, p23 and tau. However, the analysis seems to be incomplete. There are many other bands representing crosslinking products in Suppl. Fig. 8a as well, while the most of p23 and FKBP51 do not seem to engage in intermolecular crosslinking. Control experiments exploring by SDS-PAGE the crosslink products of p23 alone, FKBP51 alone, p23/tau, FKBP51/tau and FKBP51/p23 would be insightful for the interpretation of the crosslink pattern from the ternary mixture. Western blot probing against the components would allow assigning the identity of the crosslinking products.

Reply: We thank the reviewer for this suggestion. In the revised manuscript, we have analyzed another band in the SDS-PAGE gel that corresponds to the dimeric complex between FKBP51 and tau. We updated the Supplementary Fig. 8a and Supplementary Table 1 with the new data. We have also included the following sentences at the end of the result section on page no. 16:

“Apart from the band of the trimeric FKBP51-p23-tau complex, several other bands appear in the SDS-PAGE gel suggesting the presence of heterogenous complexes potentially between FKBP51-tau, tau-p23, and p23-FKBP51 (Supplementary Fig. 8a, Supplementary Table 1).”

Performing western blot probing and crosslinking reaction of p23 alone, FKBP51 alone, p23/tau, FKBP51/tau, and FKBP51/p23 for this particular study would significantly extend the timeline. Please note that we already obtained information regarding the dimeric interaction between p23-tau, FKBP51-tau, and p23-FKBP51 from NMR spectroscopy.

Were there tau-tau crosslinks also observed, i. e. is the analyzed SDS-PAGE band a complex with monomeric or oligomeric tau? Are the observed FKBP51-p23 crosslinks compatible with the model from molecular docking?

Reply: Yes, we have observed tau-tau crosslinks for both bands analyzed by MS. Yes, the observed crosslinks between FKBP51 and p23 are compatible with the structural model. We have included this in Supplementary Fig. 9:

Supplementary Fig. 9 | Mapping of the crosslinks on the structural model of the p23-FKBP51 complex. The crosslinks observed between K91 (p23) – K272 (FKBP51), K95 (p23) – K272 (FKBP51), and K95 (p23) – K385 (FKBP51) are shown, and the distances between the Nz atoms of the lysine residues indicated.

Reviewer #1:

The main comment in the last review round was on the lack of evidence for cooperativity between P23 and FKBP51 in regulating aggregation. Despite apparent acknowledgement of the comment in the response submitted, the first result is entitled “p23-FKBP51 cooperatively modulate tau aggregation”.

Figure 1 evaluate aggregation inhibition through ThT.assays. It shows no effect of FKBP51 in both ratios of tau: FKBP51. Thus no claims should be made that FKBP51 inhibit aggregation. Such claim is present for example in the abstract "Here we reveal an Hsp90-independent modulation of tau aggregation by the two Hsp90 co-chaperones p23 and FKBP51 and their molecular complex". I also don't understand the basis for the freshly added statement in the discussion “The PPlase activity of FKBP51 may enhance the aggregation of tau whereas the chaperone activity may delay tau's aggregation.”

In contrast P23 has a very significant effect on aggregation kinetics. The addition of FKBP51 at lower ratio (1:0.1:0.1) has no influence. The addition of FKBP51 to the higher ratio (1:0.2:0.2) has a small effect compare to P23. This is the only data that might point to cooperativity, but given the short statistical margin (n=3, p=0.0366) and the very small delaying effect compare to a very significant effect of P23 alone, it seems too weak to really claim a cooperativity effect. It rather looks like P23 alone is governing the aggregation delaying effect.

Reply: We thank the reviewer for the suggestions. In the revised manuscript, we have toned down all claims of cooperativity. Specifically, we made the following changes to the text –

Abstract: *‘Here we reveal an Hsp90-independent mechanism by which the co-chaperone p23 as well as a molecular complex formed by two co-chaperones, p23 and FKBP51, modulates tau aggregation.’*

The first result is now entitled – *‘p23 modulates tau aggregation alone as well as in the presence of FKBP51’*

Also, we made the following changes in the text on page 5 – *‘The combined data indicates that p23 plays a crucial role in delaying the aggregation of tau independently as well as in the presence of FKBP51.’* and *‘The findings from the in vitro aggregation assay suggest that the combined presence of FKBP51 and p23 delays tau's aggregation to a slightly larger extent than p23 alone.’*

I also don't understand the basis for the freshly added statement in the discussion “The PPlase activity of FKBP51 may enhance the aggregation of tau whereas the chaperone activity may delay tau's aggregation.”

Reply: We apologize if the written sentence gave the impression that it was derived from our experimental evidence. We based this statement on the existing literature for other PPlases like FKBP12 and CypA.

To clarify this we have added *‘We speculate that the PPlase activity of FKBP51 may enhance the aggregation of tau whereas the chaperone activity may delay tau's aggregation.’* We also added the following sentence in the discussion – *‘Thus, at the concentration of FKBP51 used in this study, the PPlase and chaperone activities might counterbalance each other, resulting in no significant effect on tau aggregation kinetics.’*

Reviewer #4:

The revised manuscript addresses most of my concerns.

The title of Fig. 7, "p23 and Hsp90 compete to form a complex with FKBP51." does not suggest that "a core element of the scheme is the protein tau", as stated by the authors in their response. Better revise to: "Competition of p23 and Hsp90 for FKBP51 modulates interactions with tau." or similar. To refer the reader to the selective view of the FKBP51/p23/Hsp90/tau interactions in Fig. 7, the legend should also state: "Of note, binary p23-Hsp90 and ternary p23-FKBP51-Hsp90 complexes do also exist (not shown)."

Reply: We thank the reviewer for the suggestions. As suggested by the reviewer, we have updated the title of Fig. 7 to '**Competition of p23 and Hsp90 for FKBP51 modulates interactions with tau**' as well as added the following sentence to the figure legend – '*Of note, binary p23-Hsp90 and ternary p23-FKBP51-Hsp90 complexes also exist (not shown).*'